# DEEP PROGRESSIVE SEARCH FOR ELECTROMAGNETIC STRUCTURE DESIGN UNDER LIMITED EVALUATION BUDGETS

## ABSTRACT

Electromagnetic structure (EMS) design aims to optimize a material distribution, *e.g.*, metals over a printed circuit board, which is crucial for antenna and meta-material. This task, however, is inherently a highly non-convex problem with no explicit objective function, making it extremely challenging to solve. The most common approach to addressing this problem relies on evolutionary algorithms (e.g., Genetic Algorithm), where candidate structures are evaluated through electromagnetic simulation using specialized software. However, these methods struggle with inefficiency, especially when dealing with large structural design space and time-consuming simulations. To address this, we propose a Deep Progressive Search method called DPS, which leverages a Deep Neural Network (DNN) as a surrogate model to identify a satisfactory structure within a limited simulation budget. Specifically, we develop a *tree-search-based design space control* strategy that models the design space as a tree and incrementally refines it through node expansions, enabling adaptive exploration of more complex regions while leveraging insights from simpler subspaces. Moreover, we introduce a *consistency-based sample selection strategy* to balance exploration and exploitation. Experiments on two real-world engineering tasks, *i.e.*, Dual-layer Frequency Selective Surface and High-gain Antenna show the effectiveness of the proposed DPS in terms of efficiency under limited evaluation budgets.

## 1 INTRODUCTION

Electromagnetic structure (EMS) is designed to interact with electromagnetic waves, which is crucial for various domains, ranging from telecommunications to 5G antennas, including frequency-selective surface (Zhu et al., 2022), metamaterials (Chen et al., 2023; Deng et al., 2021), photonic crystals (Peurifoy et al., 2018), and circuit (Cheng et al., 2022; Shahane et al., 2023). Despite its broad range of applications, EMS design is inherently challenging due to its non-convex nature and the lack of an explicit objective function. In these cases, optimization problems typically resort to evolutionary algorithms, like Genetic Algorithms, which can explore the solution space without relying on gradient information. However, such algorithms are still inefficient in solving this problem due to two major challenges.

One of the primary challenges in EMS design is the *vast problem space*. EMS optimization involves an enormous design space with $b^d$ possible candidate solutions, *e.g.*, for a $12 \times 24$ grid with each position having only two states (metal or empty), where $b = 2$, $d = 288$, leading to $2^{288} \approx 10^{86}$ kinds of possibilities. This vast problem space makes it extremely challenging for both human experts and algorithms to efficiently learn and identify effective patterns. The second major challenge is the *Costly Evaluation*. Assessing EMS designs necessitates real-time simulations that are computationally expensive, typically involving the solution of complex partial differential equations (PDEs), which cannot be substituted by simpler analytical methods (Koziel & Ogurtsov, 2014). The time required for simulating a single design ranges from 660 seconds to 42,780 seconds (see Table 1), according to a technical report from Inceptra [1]. This substantial time investment makes it infeasible to evaluate a large number of candidate designs through trial and error. Consequently, the development of efficient

---

[1] https://www.inceptra.com/how-computer-hardware-impacts-cst-electromagnetic-simulation-speed/

Figure 1: Illustration of the EMS design workflow. Subfigure (a) shows the overall workflow (Forrester et al., 2008), while Subfigures (b) and (c) depict its **Optimizer**, with (b) representing the traditional method and (c) ous.

sample selection strategies becomes essential, as they can significantly reduce the number of samples that need to be evaluated, thus lowering computational costs and speeding up the design process.

Recent advanced methods have attempted to solve the EMS design tasks, which can be broadly divided into two categories. For instance, *predictor-based methods* (Koziel et al., 2022; Jing et al., 2022) train a DNN-based predictor to replace traditional time-consuming evaluation processes and use it for sample selection; *conditional generative models based methods* (Brookes et al., 2019; Gao et al., 2023) train a generator to craft design schemes aligning with predetermined performance criteria. *Unfortunately*, these methods face the difficulty of high data-collection costs since they merely shift the time cost from the evaluation phase to the collection of the training data, rendering them of limited practical utility. Specifically, to create high-quality DNN models within a large design space, they often necessitate substantial evaluation costs to simulate enormous structures as training data. For example, Wang et al. (2023) use 10∼20 thousand data and Majorel et al. (2022) employ 20∼2, 000 thousand for training, meaning an immense time investment in the data collection phase ranging from two months to thirty years. This reliance on vast amounts of training data makes these methods impractical in real-world scenarios, where computational resources and time are limited. Therefore, reducing the evaluation costs is critical for the EMS design task.

To address these challenges, we propose a Deep Progressive Search (DPS) that focuses on design space management and sample selection. DPS introduces a **Tree-Search-based Design Space Control** (TSS) strategy, which models the design space as a tree and dynamically refines it by expanding and adjusting nodes, enabling efficient discovery of satisfactory designs within a compact search area. Coupled with the TSS, we incorporate an **Consistency-based Sample Selection** (CSS) strategy to optimize the sample evaluation process. Recognizing the challenges of unreliable model predictions, especially few-shot settings, CSS strategy dynamically adjusts the selection criteria based on the consistency of model outputs over time. This approach ensures that candidates with initially lower predicted performance are not prematurely discarded, reducing the risk of overlooking truly potential designs. By balancing exploration and exploitation, CSS enhances search efficiency and provides a more thorough evaluation of the design space. Together, these two strategies minimize the data collection and evaluation costs typically associated with EMS design, while accelerating the discovery of high-quality designs, even within the constraints of a vast and complex design space.

Our contributions are summarized as follows:

- A **Deep Progressive Search** (DPS) paradigm for efficient EMS design. We propose DPS method to reduce the high data dependency and computational costs in EMS design by optimizing both search efficiency and resource usage. By refining the exploration within a compact search space, our method significantly reduces the need for extensive simulations and large training datasets. Empirical results show that DPS not only finds high-performance solutions but also lowers evaluation costs, making it practical for large-scale EMS tasks with limited resources.

- A **Tree-Search-based Design Space Control** (TSS) strategy for progressive design exploration. Our TSS models the design space as a dynamically evolving tree. By progressively expanding and refining nodes, TSS directs the search toward more promising areas, enabling the efficient discovery of satisfactory designs. This strategy accelerates exploration and enhances the predictor's

Table 1: Comparison of different structure design tasks.

| Field | Complexity | Evaluation Method |
|---|---|---|
| DNA Sequence (Barrera et al., 2016) | $10^4 \sim 10^6$ | Dataset |
| Drug Discovery (Gaulton et al., 2012) | $10^6$ | Dataset |
| NAS (Siems et al., 2021; Dong & Yang, 2020) | $10^4 \sim 10^{18}$ | Dataset or Surrogate |
| EMS Design(ours) | $10^{86}; 10^{90}$ | Costly Simulation |

generalization across various design regions. Experiments confirm that TSS significantly reduces evaluation costs while guiding the search toward high-quality designs.

- A **Consistency-based Sample Selection** (CSS) mechanism to enhance search reliability and efficiency. Our DPS incorporates a CSS that dynamically adjusts the reliance on model predictions based on their consistency, particularly in few-shot settings. By ensuring that seemingly suboptimal candidates are not prematurely discarded, CSS maintains a balance between exploration and exploitation. This strategy reduces the interference caused by inaccurate model prediction during the search process, accelerating the identification of high-quality solutions from candidate alternatives.

## 2 RELATED WORK

### 2.1 ELECTROMAGNETIC STRUCTURE DESIGN

Advanced technique are increasingly applied to designing electromagnetic structures, notably in optimizing designs like Frequency Selective Surfaces (FSS) using surrogate models with various algorithms (Naseri et al., 2022; Jia et al., 2023; Zheng et al., 2023). However, creating accurate models in enormous design spaces poses challenges. Researchers have introduced methods like Knowledge-Based Domain-Constrained Deep Learning Surrogates (Koziel et al., 2022), which restrict the model's domain to relevant parameter regions, and MLAO-AGD (Wu et al., 2024), which updates models dynamically during searches. Additionally, generative models like cGAN (An et al., 2021) and cVAE (Lin et al., 2022) facilitate inverse design by generating compliant structures. Techniques combining generative models with heuristic algorithms, such as using VAE (Koziel et al., 2022) and Particle Swarm Optimization, enhance design stability. Moreover, Yin et al. (2024) and Yin et al. (2023) apply Monte Carlo tree search to optimize the design of wireless power transfer systems and inductors for improved efficiency and performance. However, these approaches require large datasets for high-quality models, leading to increased simulation costs.

### 2.2 ANALOGOUS STRUCTURE DESIGN

In addition to electromagnetic structure design, data-driven techniques are making significant strides in various other fields(Mirhoseini et al., 2021; Sammut et al., 2022). Notably, within the domain of protein structure prediction, a three-track network (Baek et al., 2021) has been devised by researchers to offer valuable insights into the functions of proteins with currently unknown structures. Moreover, AlphaFold (Jumper et al., 2021) merges physical and biological knowledge into protein structure by leveraging multiple sequence alignments within the framework of its deep learning algorithm. In drug discovery, neural networks predict antibacterial molecules (Stokes et al., 2020), with DrugGPS (Zhang & Liu, 2023) enhancing design through a motif-based 3D generation approach. GFlowNets (Madan et al., 2023) excel in generating diverse sequences efficiently, even in sparse or long-action environments. In chip design, RL-based models (Cheng et al., 2022; Lai et al., 2022) optimize macro placement. Notably, fields like drug discovery often rely on offline data for surrogate models, whereas tasks like neural network architecture search (Liu et al., 2018) and electromagnetic design benefit from real-time evaluation, supporting an online optimization framework that continuously updates models during design.

## 3 PROBLEM FORMULATION

In this paper, we focus on optimizing the design of EMS under limited computational resources. To formulate this problem, we first give some necessary definitions.

| **Algorithm 1** General scheme of DPS for EMS. | **Algorithm 2** EMS Optimization with TSS. |
|---|---|
| **Require:** Initial dataset $D_0$, maximum of simulation runs $T_{\max}$, maximum of tree nodes $N_{\max}$. Dataset $D \leftarrow D_0$. Current runs of simulation $T \leftarrow \text{length}(D_0)$. **while** $T \leq T_{\max}$ **do** Train initial predictor $f_\theta$. Conduct optimization under Tree-Search-based Design Space Control in algorithm 2. Conduct Consistency-based Sample Selection based on Eqn. (10). Conduct Simulation and obtain feedback $\{(x, y)\}$. Add $\{(x, y)\}$ to dataset $D$. Update predictor $f_\theta$ using $D$. Update Current runs of simulation $T$. **end while** **return** The satisfactory solutions from $\mathcal{D}$. | **Require:** Maximum of leaf nodes $N_{\max}$, predictor $f_\theta$, size $K$ of Top-K, Maximum iterations $M$. Initialize the root node $n_{\text{root}}$, leaf node set $L$ and Top-K list. **for** each $i \in [1, M]$ **do** **while** $|L| < N_{\max}$ **do** Randomly select a leaf node $n$ from $L$. Resample node state $s_n \sim \text{Uniform}(\{0, 1\})$ or split the node based on Eqn. (4). Reconstruct the design matrix $\mathbf{x}$ based on Eqn. (5) and evaluate $O(f_\theta(\mathbf{x}))$. Update the Top-K list. **end while** **end for** Conduct Depth-wise Importance Assignment based on Eqn. (7). **return** The Top-K best designs $\{\mathbf{x}_k^*\}_{k=1}^K$. |

**Design Space**: We denote a design parameter space as $\mathcal{X} \subseteq \{0, 1\}^{m \times n}$, where each sample $\mathbf{x} \in \mathcal{X}$ has an element $x_{ij}$ indicating whether some specific material is utilized in the area at the $i$-th row and $j$-th column of the electromagnetic structure.

**Performance Evaluation**: The evaluation of electromagnetic structures often involves complex electromagnetic field behaviors, involving the solution to Maxwell's equations (Bondeson et al., 2012), where this process is hard to solve analytically. To remedy this, the evaluation is usually done numerically using simulation software. We denote the simulation process as a function $S$ to map $\mathbf{x}$ from the design parameters to a $p$-dimensional vector, *i.e.*, $S(\mathbf{x}) = (S_1(\mathbf{x}), \ldots, S_k(\mathbf{x}), \ldots, S_p(\mathbf{x}))$, where each $S_k(\mathbf{x})$ represent a performance criterion corresponding to a specific performance characteristic of the electromagnetic structure. By simulating a set of sample points $\{\mathbf{x}_i\}$, we obtain a dataset $\{(\mathbf{x}_i, \mathbf{y}_i)\}$, where $\mathbf{y}_i = S(\mathbf{x}_i)$.

**Optimization Formulation**: The aim of solving EMS design problem is to maximize the performance of EMS design under limited evaluation budget, which can be characterized as a non-convex non-differentiable optimization problem. The objective function $O$ is employed to integrate multiple performance criterion, defined as $O : \mathbb{R}^p \rightarrow \mathbb{R}$, employing a linear weighted sum method, i.e., $O(S(\mathbf{x})) = \sum_{k=1}^p w_k S_k(\mathbf{x})$, where $w_k$ represents the weight of the $k$-th performance indicator. Consequently, the optimization problem for the design of electromagnetic structures is formulated as:

$$\max_{\mathbf{x} \in \mathcal{X}} O(S(\mathbf{x})) = \sum_{k=1}^p w_k S_k(\mathbf{x}), \text{ s.t. } T \leq T_{\max}, \tag{1}$$

where $T$ represents the number of simulations performed to evaluate the candidate solutions, and $T_{\max}$ is the budget of simulations. This constraint ensures that the optimization process remains feasible within the computational resources and time limits available, as extensive simulations can be both time-consuming and costly.

To reduce the simulations when optimizing (1), it is common to introduce a predictor $f_\theta$, defined by parameters $\theta$, which can approximate the simulation result $S(\mathbf{x})$. This optimization problem can thus be approximated as the following equation to accelerate the optimization process:

$$\max_{\mathbf{x} \in \mathcal{X}} O(f_\theta(\mathbf{x})). \tag{2}$$

Usually, since deep neural networks like can achieve a speedup of over 25,000 times compared to traditional simulation software (*e.g.*, 30ms vs 660s), evaluating numerous designs and selecting solutions becomes highly efficient. However, due to approximation errors, these candidates must undergo validation through high-fidelity simulations before being confirmed as satisfactory designs.

## 4 PROPOSED METHODS

In this paper, we propose a Deep Progressive Search (DPS) method, which aims to find satisfactory solutions for EMS design within the constraints of limited computational resources. We achieve

Figure 2: An illustration of the proposed DPS. The top section presents the complete flowchart of the algorithm. Given an expected objective $y^*$ and a computational budget $T_{\max}$, a Tree-search-based Design Space Control obtains candidate designs $\{\mathbf{x}\}$ within dynamical design space. Then, our Consistency-based Sample Selection exploit the consistency of the prediction to choose reliable designs for simulation. When simulated samples $\{\mathbf{x}, \mathbf{y}\}$ come, we determine whether to continue search with an expanded dataset. The lower sections provide details of specific modules: (a) our Tree-search-based Design Space Control is conducted through two stages, first performing a hierarchical tree search strategy space and then refining the designs through Depth-wise Importance Assignment; (b) Consistency-based Sample Selection uses the model's temporal prediction consistency to decide whether to adopt a conservative or greedy strategy.

this by focusing on reducing the size of design space and minimizing the ineffective utilization of knowledge due to the subpar performance of predictors. As shown in Figure 2, our DPS consists of two parts. **1)** *Tree-Search-based Design Space Control* (c.f. Section 4.1) aims to enhance the management of design space. It models the design space as a controlled search tree, allowing the model to start learning in a simple space and progressively expand to more complex spaces. **2)** *Consistency-based Sample Selection* (c.f. Section 4.2) is developed to enable the search process to accommodate a model with weaker performance. This is achieved by assessing the reliability of the model's historical predictions, which in turn guides the degree to which the model's knowledge is applied. The pseudo-code of DPS is summarized in Algorithm 1.

## 4.1 TREE-SEARCH-BASED DESIGN SPACE CONTROL

Precise management of the design space is critical for EMS design. An excessively large search space can lead to exponential growth in space complexity. Conversely, a design space that is too small limit the optimizer's ability to find satisfactory solutions. To address this challenge, we propose a method called Tree-Search-based Design Space Control (TSS). TSS consists of a Quadtree-based EMS design representation module, which manages varying resolutions across different regions through recursive subdivision, and a design space tree search module, which progressively refines the design space for efficient search and optimization. The pseudo-code of TSS is summarized in Algorithm 2.

**Representation of EMS Design Based on Quadtree.** Traditional pixel matrices apply a uniform resolution across all regions, which leads to significant redundancy when processing large regions of uniform values. Our idea thus starts from allowing for varying resolutions across different regions. To implement this, we employ a quadtree to manage and subdivide these regions. The quadtree allows simple regions to be represented by leaf nodes, while more complex regions undergo further subdivision by expanding the leaf nodes, enabling finer resolution, reducing redundancy, and ultimately enhancing representation efficiency.

Specifically, our quadtree $Q$ is a recursive structure used to provide a simplified representation for EMS design. Each node $n$ in $Q$ corresponds to a subregion of the matrix and holds values that record

the row $i$ and column $j$ ranges of the region:

$$r_n^{\text{start}} \leq i \leq r_n^{\text{end}}, \quad c_n^{\text{start}} \leq j \leq c_n^{\text{end}}, \tag{3}$$

where $r_n^{\text{start}}$ and $r_n^{\text{end}}$ represent the starting and ending rows, and $c_n^{\text{start}}$ and $c_n^{\text{end}}$ represent the starting and ending columns of the subregion. Each leaf node is a terminal node that can represents a fixed subregion of the matrix. Each leaf node has a additional value $s_n \in \{0, 1\}$, which indicates whether the corresponding subregion is entirely 0 or 1. By using this value, leaf nodes efficiently simplify the representation of matrix subregions without needing to store individual matrix elements. A leaf node can further split into four child nodes, representing the four quadrants of the region: upper-left ($n_0$), upper-right ($n_1$), lower-left ($n_2$), and lower-right ($n_3$). These child nodes recursively divide the region associated with their parent node. Each child node's row and column ranges are determined by calculating the midpoints of the parent node's ranges as follows:

$$r_{\text{mid}} = \left\lfloor \frac{r_n^{\text{start}} + r_n^{\text{end}}}{2} \right\rfloor, \quad c_{\text{mid}} = \left\lfloor \frac{c_n^{\text{start}} + c_n^{\text{end}}}{2} \right\rfloor, \tag{4}$$

Thus, the row and column ranges for the child node $n0$ are $[r_n^{\text{start}}, r_{\text{mid}}]$ and $[c_n^{\text{start}}, c_{\text{mid}}]$, respectively, with similar adjustments for the other child nodes. As nodes continue to subdivide, the quadtree grows from the root, and the division process reflects the progressive refinement of the matrix subregions. For each element $x_{i,j}$ in the EMS design matrix, its value is determined by the subregion defined by the corresponding leaf node. Therefore, the design matrix can be reconstructed as follows:

$$x_{i,j} = \sum_{n \in L} s_n \cdot \mathbb{I}_n(i, j), \tag{5}$$

where $L$ is the set of leaf nodes, and $\mathbb{I}_n(i, j)$ is an indicator function that determines whether position $(i, j)$ belongs to the subregion associated with node $n$.

The entire growth process of the quadtree proceeds by recursively subdividing the matrix regions, gradually refining the simple initial matrix into a more complex structure, with each leaf node determining whether its subregion is entirely 0 or 1. This structure efficiently compresses matrix information and manages different region resolutions via the tree.

The design space consists of all possible combinations of leaf node values. Let $L$ be the set of leaf nodes in the current quadtree, then the design space is defined as:

$$\mathcal{S} = \{\mathbf{s} = \{s_n\}_{n \in L} \mid s_n \in \{0, 1\}\}. \tag{6}$$

The size of the search space is $2^{|L|}$, where $|L|$ is the number of leaf nodes. This indicates that the complexity of the search space can be increased by expanding the leaf nodes in the quadtree. Thus, we are able to perform progressive search within a well-managed design space.

**Tree Search with Well-managed Design Space.** We propose a hierarchical search strategy, which starts from a simple design space and gradually increases the complexity of the design. By partitioning the design matrix into smaller subregions, the search process can adaptively explore finer subregions.

As shown in Figure 2(a), the tree search process begins with the root node $n_{\text{root}}$, which represents a sample in the most simplified design space. Subsequently, based on the current set of leaf nodes $L$, the design matrix $\mathbf{x}$ is reconstructed, and its performance $O(f_\theta(\mathbf{x}))$ is evaluated by a predictor $f_\theta$. The tree search process maintains the current Top-K best design matrices $\mathbf{x}_k^*$ and their corresponding performance values $O_k^*$, where $O_k^* = O(f_\theta(\mathbf{x}_k^*))$ represents the $k$-th best performance found so far.

At each iteration, a leaf node $n$ is randomly selected from $L$, and either its state $s_n$ is resampled or is split into four child nodes with randomly initialized states, with both actions following a Bernoulli distribution with parameter 0.5. Resampling explores alternative configurations without expanding the design space, while splitting increases search granularity. The design matrix $\mathbf{x}$ is reconstructed based on the results of these operations and evaluated by the predictor. If the new performance exceeds the current $k$-th best performance $O_k^*$, the Top-K list is updated, replacing the $k$-th best design matrix $\mathbf{x}_k^*$ with $\mathbf{x}$, and recording the corresponding performance value $O_k^* \leftarrow O(f_\theta(\mathbf{x}))$. The iteration repeats until the number of leaf nodes reaches the preset limit $N_{\text{max}}$, at which point the algorithm terminates and returns the final Top-K best design matrices $\mathbf{x}_k^*$.

**Depth-wise Importance Assignment**. In the process of expanding the design space, the division of nodes is initially uniform, treating each newly created leaf node as having equal importance

within the matrix. However, it is possible that some regions may have a greater impact on overall performance. Therefore, we introduce a further refinement phase for Top-k designs to better optimize the EMS design, where the design space is formulated as: In the process of expanding the design space, the initial division of nodes assumes a uniform distribution, treating each newly created leaf node as having equal importance within the EMS matrix. However, certain regions may contribute more significantly to overall performance. To address this, we introduce a further refinement phase targeting the Top-$k$ designs to better optimize the EMS structure. The optimization problem is formulated as:

$$\max_{\mathbf{s}' \in \mathcal{S}'} O(f_\theta(\mathbf{x}_{\mathbf{s}'})), \tag{7}$$

where $\mathbf{x}_{\mathbf{s}'}$ represents the EMS matrix defined by the quadtree structure, and $\mathcal{S}'$ is the search space comprising all possible partition parameters in the quadtree:

$$\mathcal{S}' = \{(r_n^{\text{start}}, r_n^{\text{end}}, c_n^{\text{start}}, c_n^{\text{end}}) \mid n \in Q\}. \tag{8}$$

### 4.2 Consistency-based Sample Selection

After the TSS module generates candidate samples, evaluating all of them at once is impractical due to the time-consuming nature of simulations. Therefore, it is crucial to prioritize the most promising samples for performance evaluation through simulation. To achieve this, we propose a **Consistency-based Sample Selection** (CSS) strategy, which optimizes the evaluation process to enhance search efficiency by dynamically adjusting the search process to accommodate predictors with moderate or even low accuracy. The core steps of this strategy are as follows.

**Ranking-based Prediction Consistency.** Optimizer prioritize ranking accuracy over prediction precision because the goal is to identify the best solution. A model that ranks solutions correctly can still guide the optimization effectively, even with imprecise predictions. Thus, we use Kendall's tau (Kendall, 1938) coefficient to directly measures the consistency of ordering results. Specifically, in each iteration, we calculate the predicted performance $O(f_{\theta_t}(\mathbf{x}))$ of the model at the current time point $t$ for candidate samples $\mathbf{x}$, and compare these values with the predicted values $O(f_{\theta_{t-1}}(\mathbf{x}))$ at the time point $t-1$. The Kendall's tau coefficient $\tau$ is calculated as follows:

$$
\begin{aligned}
\tau = \frac{2}{n(n-1)} \sum_{i<j} &\text{sign}(O(f_{\theta_t}(\mathbf{x}_i)) - O(f_{\theta_{t-1}}(\mathbf{x}_i))) \\
&* \text{sign}(O(f_{\theta_t}(\mathbf{x}_j)) - O(f_{\theta_{t-1}}(\mathbf{x}_j))),
\end{aligned} \tag{9}
$$

where $n$ is the number of data points, and $x_i, x_j$ are the candidate samples. A value of $\tau$ close to 1 indicates high consistency, while a value close to 0 or negative indicates lower consistency.

**Mixed Selection.** When the model's predictions show instability or inaccuracy, relying solely on the model's current predictions may not be the optimal choice. To mitigate the bias caused by inaccurate predictions, introducing randomness to increase exploration becomes essential. This is achieved by combining the predictor's selection with random selection. Specifically, we determine the proportion of samples selected by the predictor based on the value of Kendall's tau coefficient $\tau$. For example, if $\tau$ is 0.8, then 80% of the samples are selected by the predictor, and the remaining 20% are determined by random selection. The total number of samples $R$, with the number of best samples selected by the predictor $R_p$ and the number of samples selected randomly $R_r$, are determined as follows:

$$R_p = \tau \times R, \quad R_r = (1 - \tau) \times R. \tag{10}$$

Through this approach, the Consistency-based Sample Selection strategy effectively balances exploration and exploitation.

## 5 Experiments

We conducted a series of experiments designed to evaluate the effectiveness and robustness of our proposed method in real-world optimization tasks. The experiments answers two key questions: 1) How does our method compare to state-of-the-art approaches in terms of optimization performance and efficiency? 2) How do the individual components of our method contribute to its overall

Table 2: Detailed setting of Engineering Tasks.

| Problem | x | Design Space Dimension | $\mathbf{S}(x)$ | Objectives1 | Objectives2 |
|---------|---|------------------------|-----------------|-------------|-------------|
| DualFSS | 14*14*2 | $10^{86}$ | S-Parameters S2,1 | $\max_{\mathbf{x}} \min_{u \in [31.5,34.5]} -S(\mathbf{x})(u)$ | $\max_{\mathbf{x}} \max_{u \in [10.5,15.5]} -S(\mathbf{x})(u)$ |
| HGA | 15*20 | $10^{90}$ | Realized Gain | $\max_{\mathbf{x}} \min_{u \in [2.45,2.55]} S(\mathbf{x})(u)$ | $\max_{\mathbf{x}} \min_{u \in [5,6]} S(\mathbf{x})(u)$ |

Table 3: Comparisons on Dual-layer Frequency Selective Surface and High-gain Antenna.

| Method | Dual-layer Frequency Selective Surface | | | | High-gain Antenna | | | |
|--------|----------|-------|-------|-------------|----------|-------|-------|-------------|
| | Agg Obj ↑ | Obj1 ↑ | Obj2 ↑ | #Simulations | Agg Obj ↑ | Obj1 ↑ | Obj2 ↑ | #Simulations |
| RS | 7.2824 | 7.2824 | 36.6861 | **1000** | 0.6314 | 0.6314 | 0.7196 | **1000** |
| Surrogate-RS | 5.8116 | 5.8116 | 30.2771 | **1000** | 3.0857 | 3.0857 | 3.1845 | **1000** |
| Surrogate-GA | 4.1946 | 4.1946 | 32.1198 | **1000** | 1.5802 | 1.5802 | 4.5598 | **1000** |
| cGAN | 10.0891 | 10.0891 | **45.6102** | 7000 | -6.0131 | -6.0131 | 0.0711 | 4000 |
| cVAE | 1.1478 | 1.1478 | 28.9435 | 7000 | -3.1992 | -3.1992 | 1.2794 | 4000 |
| IDN | 4.7335 | 4.7335 | 28.8207 | 7000 | -1.7527 | **4.1657** | -1.7527 | 4000 |
| InvGrad | 2.8941 | 2.8941 | 24.6731 | 7000 | 3.1783 | 3.1783 | 4.4287 | 4000 |
| GenCO | 1.1819 | 3.9466 | 1.1819 | 7000 | -5.3032 | -5.3032 | 0.6394 | 4000 |
| DPS (Ours) | **15.1964** | **15.1964** | 31.0443 | **1000** | **3.4922** | 3.4922 | **7.7311** | **1000** |

performance? Through these investigations, we aim to demonstrate the practical advantages of our approach in handling complex real-world optimization problems. Upon acceptance of the paper, the source code will be made publicly available for further research and validation.

**Task Settings.** Our method is applied to two real-world engineering tasks: 1) Dual-layer Frequency Selective Surface (DualFSS), used for electromagnetic noise shielding around chips, and 2) High-gain Antenna (HGA), commonly used in WiFi routers, both involve two optimization objectives. Details of these tasks are provided in Table 2 and Appendix. A.

**Comparison Methods.** Our study contrasts against a variety of typical approaches, which can be broadly classified into three categories: predictor-based methods, generative methods, and random search. The predictor-based methods include: 1) Surrogate-assisted Genetic Algorithm (Surrogate-GA) (Zhu et al., 2020), which adapts a Genetic Algorithm guided by a predictor; 2) Surrogate-assisted Random Search (Surrogate-RS), which employs a random search guided by a predictor; 3) Surrogate-assisted Gradient Ascent (InvGrad) (Trabucco et al., 2022), which utilizes a predictor to acquire the gradient regarding performance with respect to design, and employs the gradient ascent optimization. The generative methods include: 1) cGAN (Generative Adversarial Network) (An et al., 2021) and 2) cVAE (Conditional Variational Autoencoder) (Lin et al., 2022), designed to generate solutions meeting specified design goals; 3) IDN (Inverse Design Network) (Ma et al., 2020), which achieves direct inverse design prediction to fulfill specified design goals by constructing reverse predictors; 4) GenCO (Ferber et al.) leverages VQ-VAE to generate structures.

**Evaluation Metrics. 1).** Aggregation Value of Objectives (Agg Obj): To evaluate the search or generation capabilities of different methods, we compare their optimal performance using the $O(S(x))$. For fair comparison, all methods use the same objective function to guide their optimization or generation process. We use the Maximin objective function to focus on maximizing the value of the worst-performing objective, ensuring a balanced optimization of multiple goals. **2).** Single Objective Value (Obj1, Obj2): Considering that structures with similar objective function values can still exhibit differences in quality. For instance, consider two solutions with objectives (10,9) and (11,9). Both have a objective value of 9. However, the second solution (11,9) is clearly superior because it performs better in one of the objectives without sacrificing the worst-case performance. Therefore, when the compared methods produce optimal results with closely matched objective function values, we continue to compare the merits of individual objectives.

**Implementation Details.** In the predictor-based approach, we employ ResNet50 as the predictor model, initialized with a dataset of 300 samples. The total number of simulation runs is limited to 1000 to maintain computational efficiency for Surrogate-GA and Surrogate-RS. For methods like Surrogate-Grad and the generative approaches (cGAN, cVAE, and IDN), which require higher model accuracy due to their more complex architectures, larger datasets are necessary. Specifically, we use 6800 and 3800 initial samples for DualFSS and HGA, resulting in final sizes of 7000 and 4000, respectively. More implementation details are in Appendix. B.

Table 4: Effectiveness of Number of Variables $N_{max}$.

| $N$ | Agg Obj↑ | Obj1(dB)↑ | Obj2(dB)↑ | Kendall's Tau↑ |
|---|---|---|---|---|
| 16 | 3.0867 | 3.0867 | 6.0295 | **0.2838 ± 0.0527** |
| 32 | **3.4922** | **3.4922** | **7.7311** | 0.2324 ± 0.0523 |
| 64 | 3.0233 | 3.0233 | 5.1847 | 0.1255 ± 0.0297 |

Table 5: Effectiveness of TSS and CSS.

| TSS | CSS | Agg Obj↑ | Obj1(dB)↑ | Obj2(dB)↑ |
|---|---|---|---|---|
| | | 3.0857 | 3.0857 | 3.1845 |
| ✓ | | 3.2209 | 3.2209 | 6.3369 |
| ✓ | ✓ | **3.4922** | **3.4922** | **7.7311** |

## 5.1 COMPARISONS ON DUAL-LAYER FREQUENCY SELECTIVE SURFACE

We report the comparisons on Dual-layer Frequency Selective Surface in Table 3. Our approach outperforms every baseline methods by a wide margin on aggregation of objective value. It is worth-noting that the objective values of predictor-based methods are significantly worse than random search under the few-shot setting of 1000 samples. This confirms that in few-sample situations, the inferior performance of predictor models indeed affects search capabilities. In contrast, our method achieved a 109% improvement in the objective value under the same simulation cost, substantially enhancing the optimization capability.

On the other hand, generative models struggle to complete training with only 1000 simulation samples, often leading to model collapse. Even when the number of simulation samples is increased to 7000, among all generative methods, only cGAN exceeds random search, yet it still falls short compared to our DPS. This reveals that, in comparison to existing generative methods, DPS not only reduces simulation costs by a factor of seven but also improves performance by at least 50.6%.

## 5.2 COMPARISONS ON HIGH-GAIN ANTENNA

Table 3 presents comparisons on High-Gain Antenna. Our DPS method outperforms the baseline, requiring only 1000 samples to achieve an objective function value of 3.49dB, with gains of 3.49dB and 7.73dB in two WiFi bands. While InvGrad and IDN perform better than Random Search, they demand more simulations and fail to meet both objectives for dual-band router antenna design.

The results show that DPS excels across various real-world tasks. Its success is due to TSS, which enhances structural feature learning for higher prediction accuracy with small samples, and CSS, which balances exploration and exploitation. In contrast, existing methods struggle with limited data, hindering their ability to capture the full problem space and achieve cost-efficient designs.

## 5.3 ABLATION STUDIES

**Study on the Number of Variables**. As outlined in Section 4.1, the variable $N_{max}$ shapes the complexity of the design space and directly influences the challenge of identifying high-quality samples. In this section, we examine the impact of varying $N_{max}$ in the context of the HGA task detailed in the Implementation Details section. We evaluate $N_{max}$ values of 16, 32, 64, and as shown in Table 4, our DPS achieves satisfactory performance with an objective function value of 3.49 dB when $N_{max} = 32$. It indicates that a smaller $N_{max}$ is prone to trapping the search in locally sub-optimal regions, while a larger $N_{max}$ makes it difficult to construct accurate models given limited computational resources.

We computed the mean and variance of Kendall's Tau (KTau) for predictors trained on samples generated with varying $N_{max}$ values, with a fixed total sample size of 1000. Samples were split into validation, test, and training sets, and hyperparameter tuning was performed separately for each dataset. Using optimal parameters, we conducted 10 trials per predictor and calculated KTau. Results showed that $N_{max}$ significantly affects KTau, with $N_{max} = 16$ yielding the best performance. As $N_{max}$ increased, KTau decreased, indicating that predictors map the design space more accurately when it is smaller. This supports TSS, showing that expanding from low to high-dimensional spaces enhances predictor performance with limited samples, improving optimization reliability.

**Effectiveness of Tree-Search-based Design Space Control**. We conduct experiments to further demonstrate the effectiveness of our progressive space design. Specifically, we compare our methods with a variant which replaces progressive design strategy with random sampling. Our experiments are conducted in High-gain Antenna under the same computational budget and the results are reported in Table 5. From the results, our method outperform the variant without progressive search, generating EMS design with higher objective function value (e.g., 3.22$dB$ vs 3.09$dB$). These results demonstrate the necessity of the proposed progressive search.

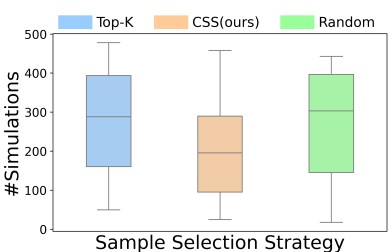

RS      Surrogate-GA      cGAN      DPS(ours)

Figure 5: Satisfactory Electromagnetic Structures of Different Methods on the High-gain Antenna.

**Effectiveness of Consistency-based Sample Selection**. To verify the effectiveness of the proposed CSS strategy, we compare our methods with a variant that simply select the top-$M$ structures evaluated by the predictor. Our experiments are conducted in High-gain Antenna and we present the objective value in Table 5.It is evident that our method outperforms the variant without CSS, yielding EMS design with higher objective function values (e.g., 3.22 $dB$ vs. 3.09 $dB$). These results show that our method could alleviate the bias induced by inaccurate predictions.

To more clearly demonstrate the advantages of our method, we designed an additional experiments. A dataset of 1000 samples was randomly divided into two equal parts. The first part was used as the initial training for the predictor.The labels of the second part were masked. We applied three different sample selection strategies—CSS (ours), Top-K, and Random—to identify the actual optimal solution within the set. In each round, 20 samples were selected and added to the training set to update the predictor, with this process continuing until the optimal sample was found. Each experiment was repeated 20 times. The results in Figure 3 demonstrate that our method outperforms traditional approaches, improving search efficiency by 50%.

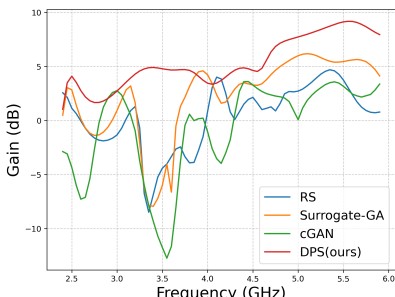

Figure 3: Simulation Costs for Optimal Solution Across Sample Selection Strategy.

### 5.4 VISUALIZATIONS

**Visualizations of Electromagnetic Structures on High-gain Antennas**. We illustrate the satisfactory electromagnetic structures from our proposed DPS and some of the compared methods. More visualizations are presented in Appendix. C. As illustrated in Figure 5, the structures generated by DPS are more regular and better aligned with the practical manufacturing constraints of engineering applications. By avoiding fragmented designs that are difficult to fabricate in real-world settings, our approach ensures improved manufacturability, thereby enhancing its practicality for engineering implementation. Besides, the results indicate that the satisfactory structures obtained through our proposed method exhibit higher values in the frequency ranges [2.45, 2.55] and [5.00, 6.00]. This implies that our design demonstrates superior performance, providing validation for the effectiveness of our algorithm.

Figure 4: Simulation Results for Satisfactory Electromagnetic Structures of Different Methods on the High-gain Antenna.

## 6 CONCLUSION

In this paper, we propose a Deep Progressive Search method under Limited Data. Specifically, we devise a Tree-Search-based Design Space Control method. By progressively searching in the simplified space, the quality of samples is improved, thus reducing dependence on the number of training samples. In addition, we introduce a Consistency-based Sample Selection. With this strategy, the search process can achieve a better balance between exploration and exploitation. Extensive experimental results on real-world engineering tasks demonstrate the effectiveness of our method.

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

# APPENDIX

In the supplementary, we provide more implementation details and more experimental results of our DPS. We organize our supplementary as follows.

- In Section A, we provide more details of the considered two real-world challenging electromagnetic structures, dual-layer frequency selective surface and high-gain antenna

- In Section B, we depict more implementation details of our DPS and the compared methods.

- In Section C, we give more experimental results to demonstrate the effectiveness of our DPS.

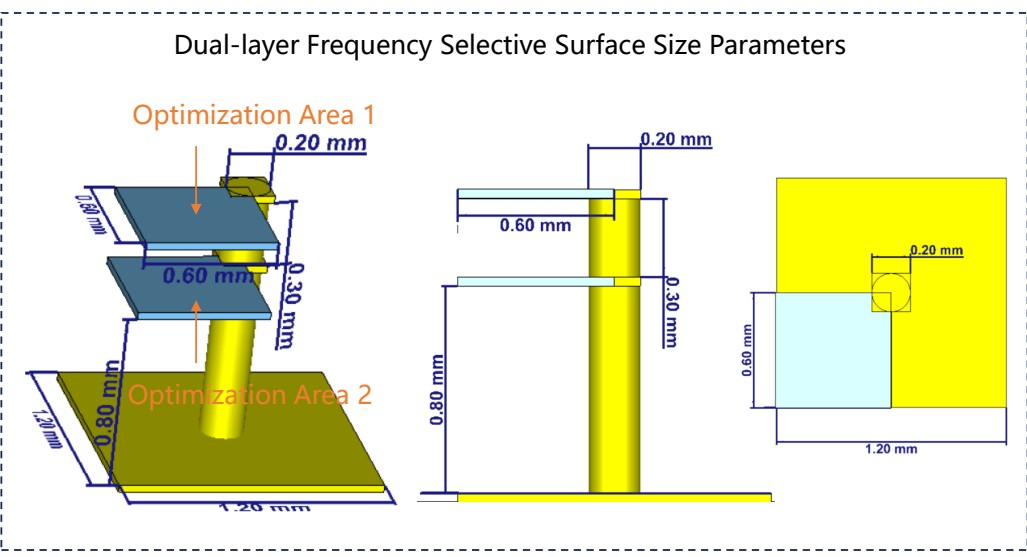

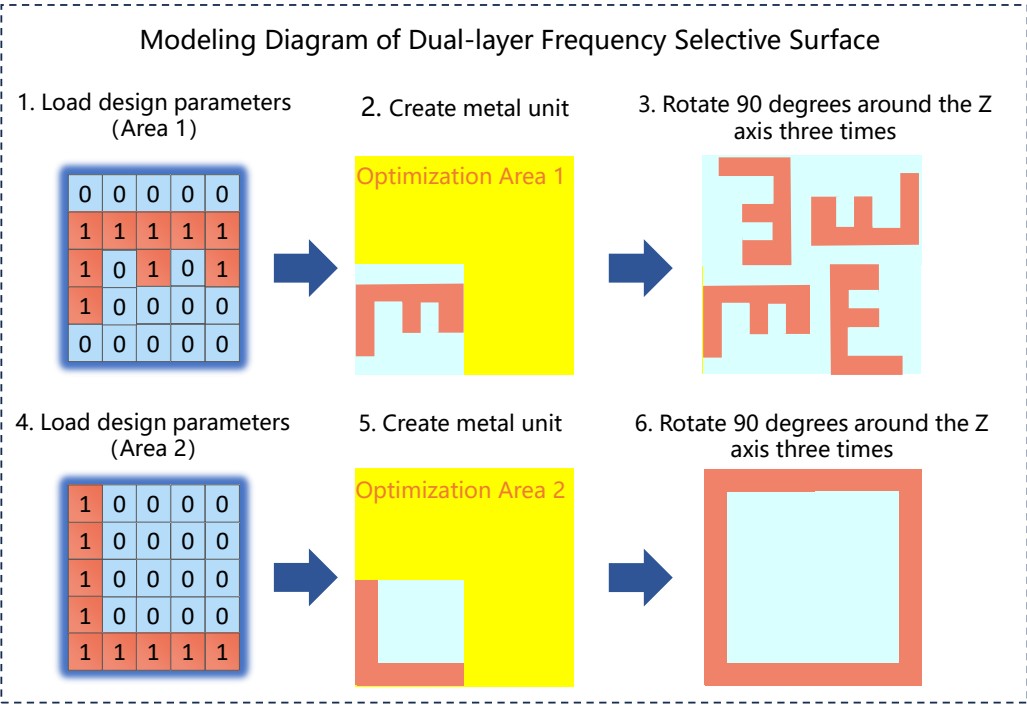

Figure 6: The detailed settings of the Dual-layer Frequency Selective Surface.

# A MORE DETAILS ON ELECTROMAGNETIC STRUCTURES

**Dual-layer Frequency Selective Surface**. The dual-layer Frequency Selective Surface (DualFSS) is an electromagnetic structure specifically designed for selectively filtering electromagnetic waves. It consists of two layers of conductive elements, each containing a grid or array of metallic elements exhibiting specific resonant behavior at particular frequencies. This configuration endows the DualFSS with frequency-dependent transmission and reflection characteristics. Despite its more intricate structure compared to a single-layer FSS, the DualFSS provides higher degrees of freedom, allowing for more flexible performance adjustments across different frequencies. In practical applications, the DualFSS finds common usage in scenarios demanding enhanced performance and broader frequency coverage, such as RF communication, radar systems, and engineering designs within the radio frequency spectrum. In our experiments, we utilized the proposed methodology to design the structure of the DualFSS, focusing on the two layers of metallic grids. The designed structure aims to meet specific performance criteria, with detailed parameters and expected metrics outlined below.

Specifically, the DualFSS under investigation is depicted in Figure 6. The FSS is composed of three layers of boards, each having a thickness of 0.035mm, and features cylindrical elements with a radius of 0.1mm. It is noteworthy that the bottommost layer, representing the grounded metallic surface, remains unaltered throughout the optimization process. In contrast, the upper two layers, initially comprising entirely of air, constitute the optimization space. The objective of the optimization is to strategically convert certain regions within the air layers into metallic elements, adhering to the constraint that each designed metallic block must have a minimum size of 0.2*0.2mm. The optimization encompasses determining the specific configuration of metallic blocks within the upper layers to achieve desired electromagnetic properties. Importantly, the maximum extent of the air region available for optimization corresponds to the footprint of the bottommost layer.

In terms of optimization objectives, this scenario aims to eliminate electromagnetic noise caused by ultra-high-frequency circuits, preventing such noise from interfering with the operation of mobile phone cameras. The high-frequency noise primarily occurs in two frequency bands: 10.5–15.5 GHz and 31.5–34.5 GHz. We evaluate the suppression capability using the S-Parameter S2,1, which is a parameter that takes only negative values. A smaller magnitude of this parameter indicates stronger suppression performance. We aim to achieve broad absorption capability in the 31.5–34.5 GHz range, while maximizing suppression in the 10.5–15.5 GHz range. To enhance generalization, it is necessary to minimize the maximum value within the 31.5–34.5 GHz band, while minimizing the minimum value within the 10.5–15.5 GHz band to strengthen absorption performance. Accordingly, we define the first objective as minimizing the maximum value of S-Parameters S2,1 over the 31.5–34.5 GHz band, represented by the formula $\min_{\mathbf{x}} \max_{u \in [31.5, 34.5]} S(\mathbf{x})(u)$. By taking the inverse, we can transform it into a maximization problem and define it using a new mathematical expression $\max_{\mathbf{x}} \min_{u \in [31.5, 34.5]} -S(\mathbf{x})(u)$, and denote this as Obj1. Similarly, the second objective is to minimize the minimum value of S-Parameters S2,1 over the 10.5–15.5 GHz band, defined by the formula $\min_{\mathbf{x}} \min_{u \in [10.5, 15.5]} S(\mathbf{x})(u)$. We can also transform it into a maximization problem and represented by the formula $\max_{\mathbf{x}} \max_{u \in [10.5, 15.5]} -S(\mathbf{x})(u)$, and referred to as Obj2. In this formulation, $\mathbf{x}$ represents the vector of structural design parameters, while $u$ denotes the frequency in GHz, serving as the independent variable across both frequency bands. The term $S(\mathbf{x})(u)$ refers to the S-Parameters S2,1 of the FSS for a given design $\mathbf{x}$ at a specific frequency $t$.

An aggressive objective function is set to maximize the worst-case performance of the single objective to achieve balanced shielding capabilities. This is mathematically expressed as $\max_{\mathbf{x}} (\min(Obj_1(\mathbf{x}), Obj_2(\mathbf{x})))$, where $Obj_1(\mathbf{x})$ represents $\min_{u \in [31.5, 34.5]} -S(\mathbf{x})(u)$ while $Obj_2(\mathbf{x})$ represents $\max_{u \in [10.5, 15.5]} -S(\mathbf{x})(u)$.

**High-gain Antenna**. The high-gain antenna is a specialized electromagnetic structure designed to achieve significant directional amplification of radio frequency signals. This type of antenna is characterized by its ability to focus transmitted or received signals in a specific direction, resulting in a concentrated radiation pattern. In the design of it, the integration of a metal array plays a crucial role in shaping the antenna's radiation pattern and achieving enhanced performance. The metal array structure involves a carefully arranged grid or array of metallic elements, such as reflectors and directors, strategically positioned to optimize the antenna's gain and directional characteristics. High-gain antennas find extensive applications in scenarios requiring long-range communication, satellite communication, and situations where a concentrated signal strength is essential.

Specifically, our structure is a rectangular prism with dimensions of 60 mm in length, 40 mm in width, and 4.6 mm in thickness in Figure 7. The prism is then divided into two halves along the midpoint of its length, parallel to the width. One half is designated as the design region, and after the design is completed, it is mirrored across the vertical plane of symmetry.

The target of this scenario is to design a dual-band router antenna operating at both 2.4 GHz and 5 GHz, ensuring optimal communication performance in both frequency bands simultaneously, rather than having one band perform well while the other lags. Consequently, our two objective functions represent the minimum Realized Gain for the 2.4 GHz and 5 GHz bands, respectively. An Aggregation Value of Objective is set to maximize the worst-case performance across both objectives, aiming to achieve effective dual-band communication.

Specifically, since the minimum value within a given range dictates the weakest communication capability of that band, to meet the communication requirements of the 2.4 GHz band, the first objective is defined as maximizing the minimum Realized Gain in the 2.45–2.55 GHz frequency range. This is mathematically expressed as $\max_{\mathbf{x}} \min_{u \in [2.45, 2.55]} S(\mathbf{x})(u)$ and denoted as Obj1. In this formulation, $x$ represents the vector of structure design, while $u$ denotes the frequency in GHz, acting as the independent variable across both frequency bands. The term $S(\mathbf{x})(u)$ denotes the Realized Gain of the antenna for a given design $x$ at a specific frequency $u$. Similarly, to ensure robust communication in the 5 GHz band, the second objective is defined as maximizing the minimum Realized Gain in the 5.0–6.0 GHz frequency range, expressed as $\max_{\mathbf{x}} \min_{u \in [5,6]} S(\mathbf{x})(u)$ and denoted as Obj2.

To achieve strong communication performance across both bands, the smaller of the two objective values is chosen as the Aggregation Value of Objective. This is mathematically expressed as $\max_{\mathbf{x}} \min \left( \min_{u \in [2.45, 2.55]} S(\mathbf{x})(u), \min_{u \in [5,6]} S(\mathbf{x})(u) \right)$. This formulation ensures that the antenna design is optimized for both frequency bands by focusing on improving the worst-case communication performance across the two bands, thereby achieving balanced and robust dual-band communication.

# B  MORE IMPLEMENTATION DETAILS

**DPS (ours).** For the proposed DPS, we maintain a consistent setup across both experimental scenarios. In both cases, the initial dataset comprises 300 samples, derived from a progressive design strategy, which allows for more adaptive and efficient sampling. The design variable $N_{max}$ is set to 32. This consistency across scenarios highlights the flexibility and robustness of DPS, as it maintains performance while adapting to different experimental conditions. Additionally, we also utilize ResNet50 with the same network architecture as the predictor, ensuring consistency in model structure across different approaches and facilitating a fair comparison of performance. We set maximum iteration $M = 10000000$ and $K = 10$. In CSS, the total number of samples $R$ is 10.

**Compared Methods.** We re-implement the following state-of-the-art electromagnetic structures design methods in our two challenging task, dual-layer frequency selective surface and high-gain antenna. More details of the baseline design methods are provided in the subsequent discussion.

- **Random Search (RS)**. In this approach, we randomly generate 1000 electromagnetic structures, evaluating them with the simulation software.

- **Surrogate-assisted Random Search (SRS)**. This approach adopts the ResNet50 architecture as the surrogate model in both scenarios, consistent with the setup in DPS, to ensure a fair comparison between the methods. The surrogate model is designed to provide prediction of the objective function given the input of the electromagnetic structures and is ultilized to guide the random search. Specifically, we begin by randomly sampling 300 electromagnetic structures to train an initial surrogate model. In each subsequent iteration, the surrogate model guides the selection of the Top-K samples from M randomly sampled candidates (where $M \gg K$), which are then evaluated through simulation. The surrogate model is updated accordingly, and this process is repeated until the total number of simulated samples reaches the predefined limit of 1000. Finally, we select the best one as optimized result. In practice, we set $M = 200000$ and $K = 10$ for our experiments in both two real-world tasks.

- **Surrogate-GA** (Zhu et al., 2020). This method exploit a surrogate model to accelerate the evolutionary algorithm. Specifically, this method fit a DNN-based surrogate model with a simulated

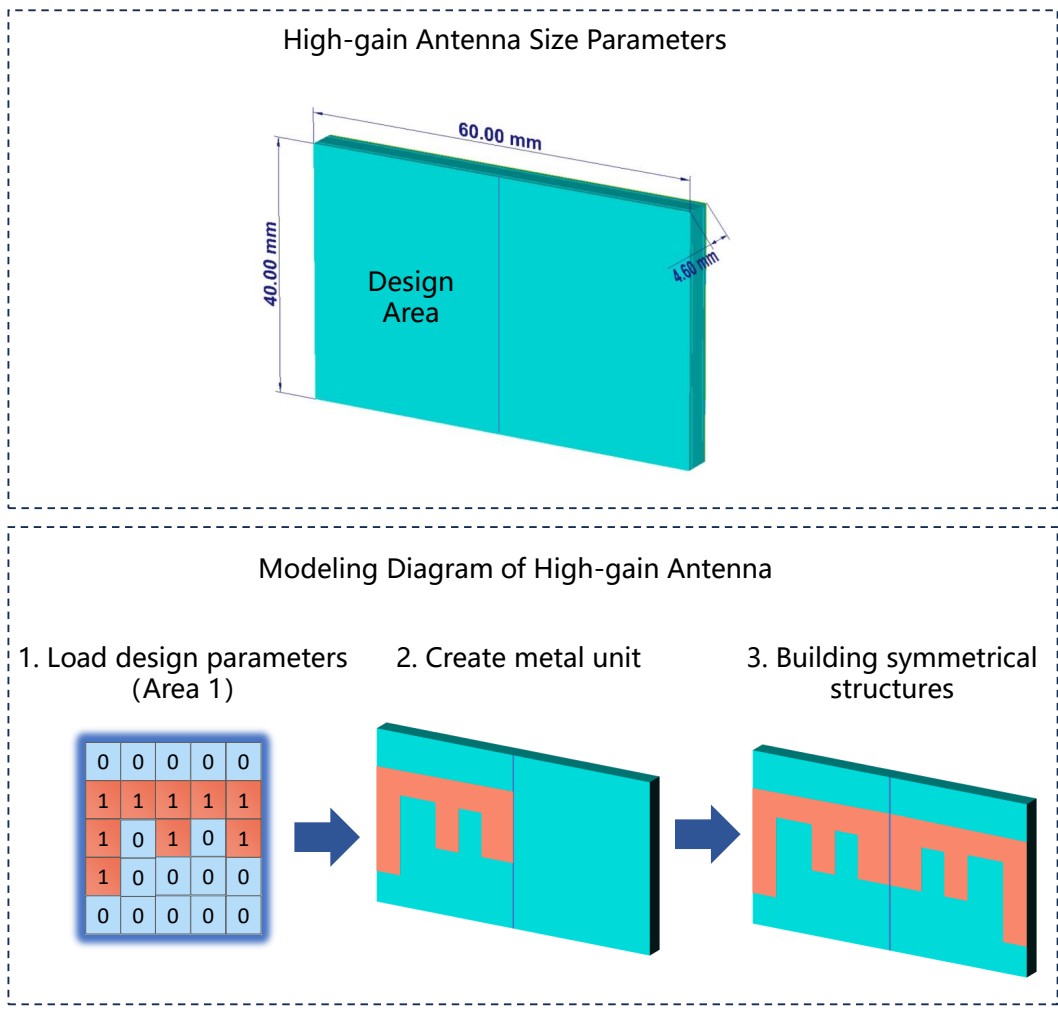

Figure 7: Designs of the High-gain Antenna.

dataset to assist fitness evaluation during the evolution process. In particular, we have adapted the mutation operator to suit our electromagnetic structural scenario by modifying it to perform transformations between 0 and 1 in the matrix elements. In our experiments, we use the ResNet50 model as the surrogate model. The model's batch size is set to 256, trained for 200 epochs, with a learning rate of 0.01. First, 300 samples were obtained through random sampling, and simulations were performed on these samples to train the surrogate model. Based on the surrogate model, we set $K$=10 for our experiments, meaning that in each generation, the top 10 samples are selected using a Top-K strategy for simulation verification. The surrogate model is then updated with these results, and a new population is generated. This process continues until the total number of simulated samples reaches 1000.

- **InvGrad** (Trabucco et al., 2022). Trabucco et al. introduces a simple baseline method based on gradient ascent. In this approach, a ResNet50-based surrogate model is initially trained on a dataset of electromagnetic structures, which is designed to establish an accurate mapping between the electromagnetic structure and the objective function. Specifically, the dataset consists of 6800 randomly sampled simulation samples for DualFSS and 3800 for HGA, which represent the minimum number of samples required to ensure the stability and reliability of the model without compromising its performance. Subsequently, the method performs multiple gradient updates on the input electromagnetic structure based on the surrogate model output, ultimately yielding an satisfactory electromagnetic structure that satisfies the specified criteria. The gradient update could be formulated as $x_{t+1} \leftarrow x_t + \alpha \nabla_x f(x)$, where $t$ represents the update step and $\alpha$ denotes the learning rate. In pratice, we set $T = 1000$, $\alpha = 0.01$ for two design tasks. We randomly sampled 10 electromagnetic structures, input them into the surrogate model for optimization, and forwarded

the optimized results to the simulation software for evaluation. The surrogate model is then updated with these results, and this process is repeated until the total number of simulated samples reaches the predefined limit of 7000 for DualFSS and 4000 for HGA.

- **IDN** (Ma et al., 2020). Ma et al. introduces a baseline method for inverse design based on Convolutional Autoencoder Network (CAN) and Inverse Design Network (IDN). In this approach, the authors utilized CAN to compresses input spectrums with the dimension of 1*1000 into low-dimensional spectrums with the dimensionof 1*50. Subsequently, the compressed latent space values were fed into the IDN with the expectation of generating structures that conform to the input spectrum. In our experiments, we set the target values as either the maximum or minimum values within a specific frequency range, eliminating the involvement of high-dimensional inputs. Consequently, we exclusively adopted the IDN component of the method for our purposes.In terms of network architecture, we introduced an additional fully connected layer before the first convolutional layer of the Inverse Design Network. This layer elevates the input target values to a 50-dimensional space to align with subsequent dimensions. The model's batch size is set to 128, trained for 200 epochs, with a learning rate of 0.0001. Adam optimizer is employed, and MAE(Mean Absolute Error) is used for loss computation. Initially, we performed random sampling to generate 6800 samples for DualFSS and 3800 samples for HGA. These samples were simulated to calculate their respective objective values, which were subsequently used to train the initial models. In the next stage, the models were utilized to generate 10 additional samples, which underwent simulation-based validation. The validated samples were then added to the dataset, and the models were retrained iteratively. This process was repeated until the simulation budget of 7000 and 4000 was reached for DualFSS and HGA, respectively. Ultimately, the sample with the best simulation performance during this process was selected as the final result.

- **cGAN** (An et al., 2021) An et al. presents a generative adversarial network that can generate metasurface designs to meet design goals . Generative adversarial nets can be extended to a conditional model if both the generator and discriminator are conditioned on some extra information. It could be any kind of auxiliary information, such as class labels or data from other modalities. We can perform the conditioning by feeding extra information into the both the discriminator and generator as additional input layer. Consequently, cGAN introduces extra information as conditions in both the encoder and decoder inputs to confer the ability to generate pecific structures based on varying conditions.The model's batch size is set to 64, trained for 200 epochs, with a discriminator learning rate of 0.00005 and generator learning rate of 0.0002. In addition, the latent dimension is set to 100, Adam optimizer is employed. We began by randomly sampling 6800 instances for DualFSS and 3800 instances for HGA, followed by simulations to derive their objective values for training the initial models. Using these models, 10 new samples were generated and validated through simulations. These validated samples were incorporated into the dataset, and the models were updated iteratively. This iterative procedure continued until the simulation budgets—7000 for DualFSS and 4000 for HGA—were exhausted. The final result was determined by selecting the sample exhibiting the optimal performance during simulations.

- **cVAE** (Lin et al., 2022). Lin et al. introduces an approach utilizing Conditional Variational Autoencoder (cVAE) to generate metasurface retroreflectors (MRF) structures satisfying specified performance criteria. cVAE represents a variant incorporating both Variational Autoencoder (VAE) and Autoencoder (AE) principles. While VAE extends the encoding-decoding training paradigm of AE by transforming it from encoding inputs into a single point in latent space to encoding inputs into a distribution in latent space, endowing it with generative capabilities, the generated content is inherently uncontrollable. Consequently, cVAE introduces conditions in both the encoder and decoder inputs to confer the ability to generate specific structures based on varying conditions.The model's batch size is set to 128, trained for 200 epochs, with a learning rate of 0.0005. In addition, the latent dimension is set to 20, Adam optimizer is employed, and the loss function is obtained through linear summation of Mean Squared Error (MSE) and 0.00000001 times the Kullback-Leibler (KL) divergence. An initial random sampling of 6800 samples for DualFSS and 3800 samples for HGA was conducted, with simulations performed to compute the corresponding objective values for initial model training. The trained models then produced 10 new samples, which were subjected to simulation validation. These validated samples were appended to the dataset, and the models were retrained iteratively until the simulation budgets of 7000 for DualFSS and 4000 for HGA were fully utilized. The final output was chosen as the sample demonstrating the highest simulation performance during the process.

Table 6: Effect of Importance Assignment.

| Method | Objective Function Value↑ | Objective1(dB)↑ | Objective2(dB)↑ |
|---|---|---|---|
| DPS w/o IA | 2.32 | 2.32 | 5.57 |
| DPS | **3.49** | **3.49** | **7.73** |

- **GenCO** (Ferber et al.) GenCO utilizes VQ-VAE (Variational Quantized Autoencoders) to generate a variety of designs that account for specific constraints, such as those encountered in nanophotonic materials. Following the approach outlined in the original paper, we integrate electromagnetic structure performance as a constraint objective into the model's training loss function. GenCO requires computing the gradient of the objective function with respect to the design variables. However, in our case, obtaining such gradient information directly through simulation is unavailable. To overcome this challenge, we use a surrogate model to approximate the gradient. The surrogate model, implemented using ResNet50, serves to predict the objective function's gradient efficiently. The training parameters of the surrogate model, such as the network architecture and optimization procedure, are consistent with those used in similar works. Since the original paper does not provide detailed hyperparameter settings for VQ-VAE, we made reasonable choices based on standard practices for training generative models. We use a four-layer convolutional and transposed convolutional network architecture for the VQ-VAE model. The specific training parameters for our implementation are as follows: latent dimension = 256, number of embeddings = 512, learning rate = 1e-3, and the number of epochs = 100.

## C MORE EXPERIMENTAL RESULTS

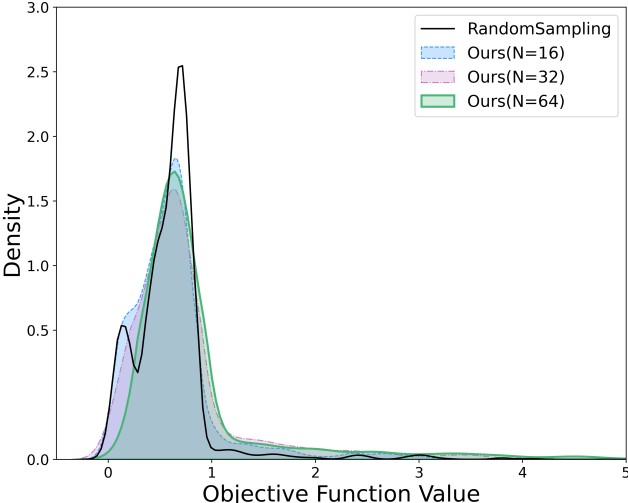

Figure 8: Comparison of Sample Performance Distribution under Different Parameter $N$ Settings.

**Effect of the Number of Variables** $N_{max}$. To further illustrate the effect of the parameter $N_{max}$, we present kernel density estimation (KDE) plots of the sample performance distribution under different parameter settings in Figure 8. We sampled and simulated 1000 samples for each parameter setting and random sampling. The experimental results demonstrate that our method is more likely to sample higher-performing instances across various parameter configurations, whereas most of the samples generated through random sampling tend to cluster in the lower performance range.

**Effect of Depth-wise Importance Assignment**. We investigate the effect of the depth-wise importance assignment. For a fair comparison, we conduct this experiment under the same simulation budget in high-gain antenna design task. From Table 6, without importance assignment, the DPS tends to find sub-optimal electromagnetic structure. When equipped with the proposed importance assignment, the searched structure consistently outperforms that without importance assignment (e.g., $3.49dB$ vs $2.32dB$). These findings illustrate the essential nature and efficacy of the introduced depth-wise importance assignment.

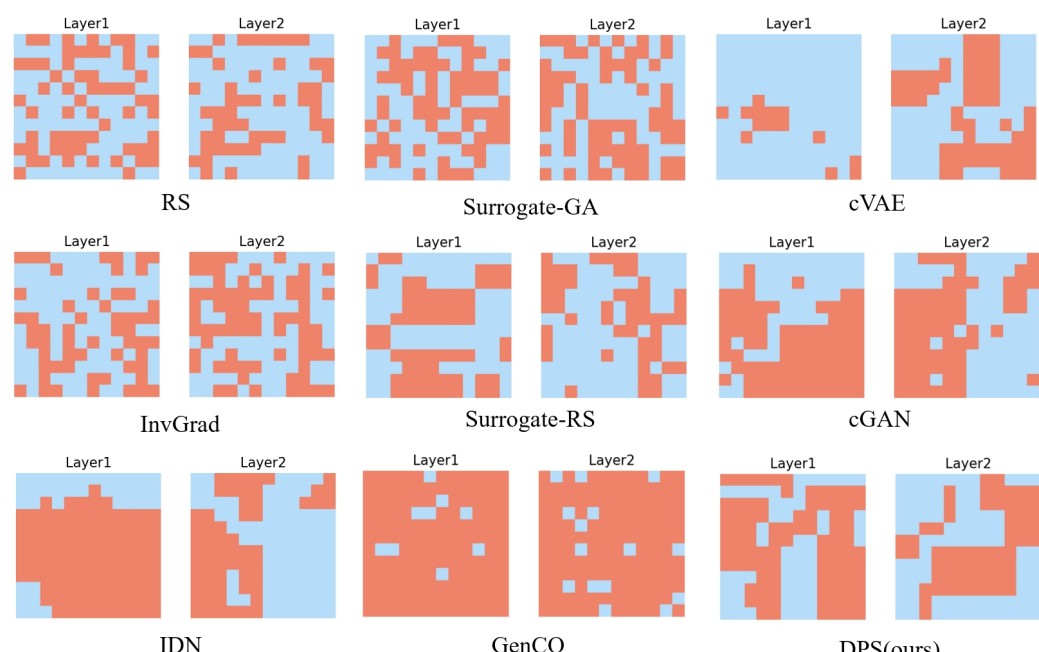

Figure 9: Satisfactory Electromagnetic Structures of Different Methods on Dual-layer Frequency Selective Surface.

**Comparisons of electromagnetic structures on Dual-layer Frequency Selective Surface**. In Figure 9, we visualize the satisfactory electromagnetic structures searched by our proposed method and the baseline methods in the task of dual-layer frequency selective surface. It can be observed that, compared to the baseline methods, the structures designed by our approach exhibit a better adherence to physical priors, showcasing a more regular and manufacturable design.

**Comparisons of Simulated Results on Dual-layer Frequency Selective Surface**. In Figure 10, We present the simulated results of the optimize electromagnetic structures for our methods and all baseline methods. From the results, it is evident that the satisfactory electromagnetic structures obtained through our proposed method exhibit lower values in the frequency ranges [31.5, 34.5] and [10.5, 15.5]. This indicates that our dual-layer frequency-selective surface performs better, providing empirical evidence for the effectiveness of our optimization algorithm.

**Comparisons of electromagnetic structures on High-gain Antenna**. In Figure 11, we visualize the satisfactory electromagnetic structures designed by our proposed method and the baseline methods in the task of high-gain antenna. It can be observed that, compared to the baseline methods, the structures designed by our approach also exhibit a better adherence to physical priors, showcasing a more regular and manufacturable design. This further demonstrates the strong generalization ability and robustness of our proposed method, proving its effectiveness across multiple real-world tasks.

**Comparisons of Simulated Results on High-gain Antenna**. In Figure 12, We further present the simulated results of the satisfactory electromagnetic structures for our methods and all baseline methods. The experiments further demonstrate that in the frequency ranges [2.45, 2.55] and [5.00, 6.00], the structures designed by our method significantly outperform all baseline methods, achieving substantial performance improvements in the target frequency bands.

**Visualizations of Satisfactory Dual-layer Frequency Selective Surface**.We illustrate the satisfactory electromagnetic structures obtained through our proposed methodology and the reference methods in the High-gain Antenna task. In Figure 13, in contrast to the reference methods, the structures formulated by our approach demonstrate a heightened conformity to physical priors, presenting a more regular and manufacturable design.

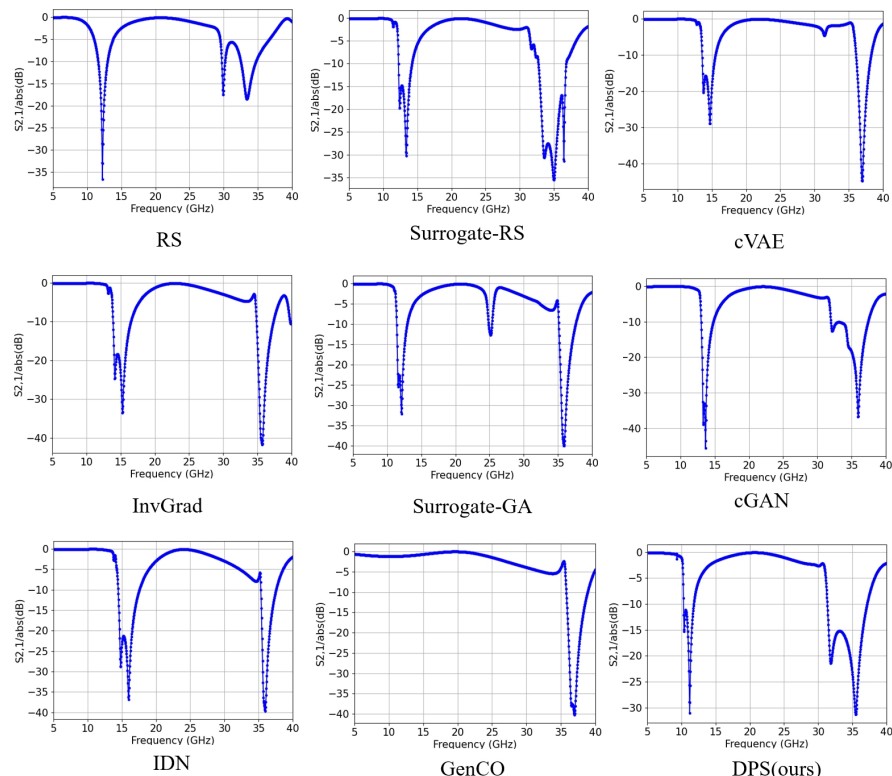

Figure 10: Simulated Results of Satisfactory Electromagnetic Structures on Dual-layer Frequency Selective Surface.

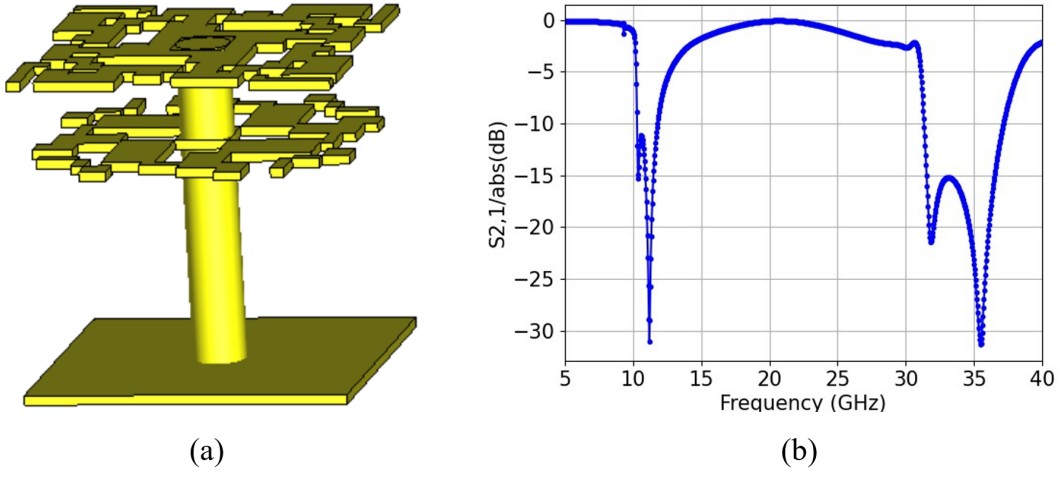

(a)                                                              (b)

Figure 13: (a) Satisfactory Electromagnetic Structures of Different Methods on Dual-layer Frequency Selective Surface. (b) Simulated Results of Satisfactory Electromagnetic Structures on Dual-layer Frequency Selective Surface.

**Visualizations of Satisfactory High-gain Antenna**. In Figure 14, the results indicate that the satisfactory electromagnetic structures obtained through our proposed method exhibit higher values in the frequency ranges [2.45, 2.55] and [5.00, 6.00]. This implies that our high-gain antenna demonstrates superior performance, providing validation for the effectiveness of our algorithm.

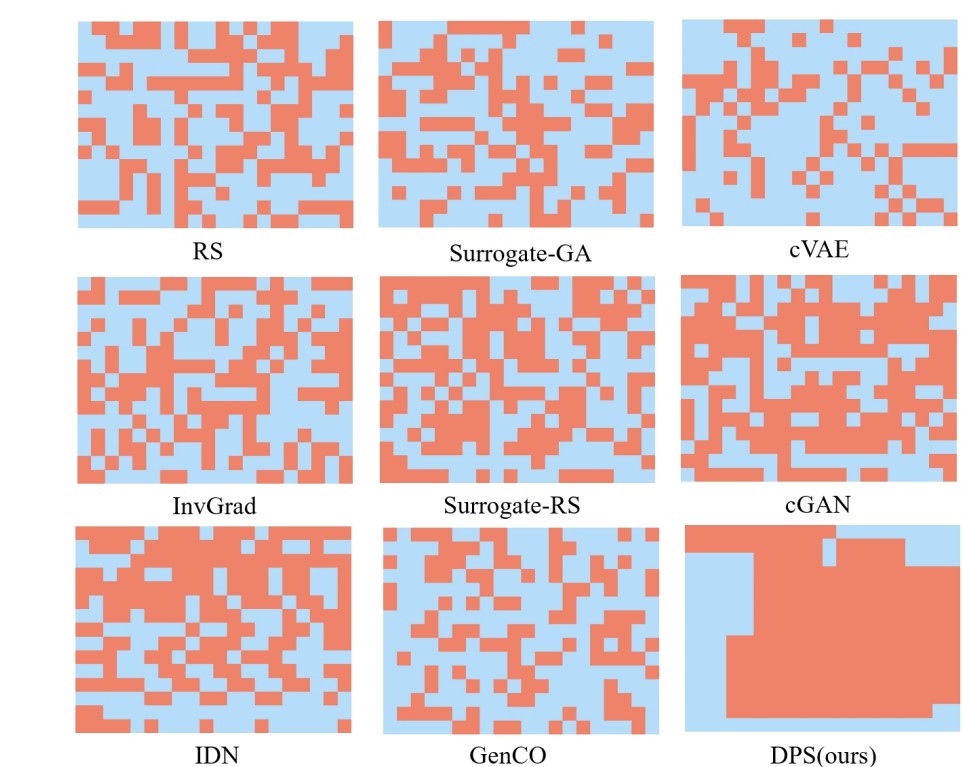

Figure 11: Satisfactory Electromagnetic Structures of Different Methods on Dual-layer Frequency Selective Surface.

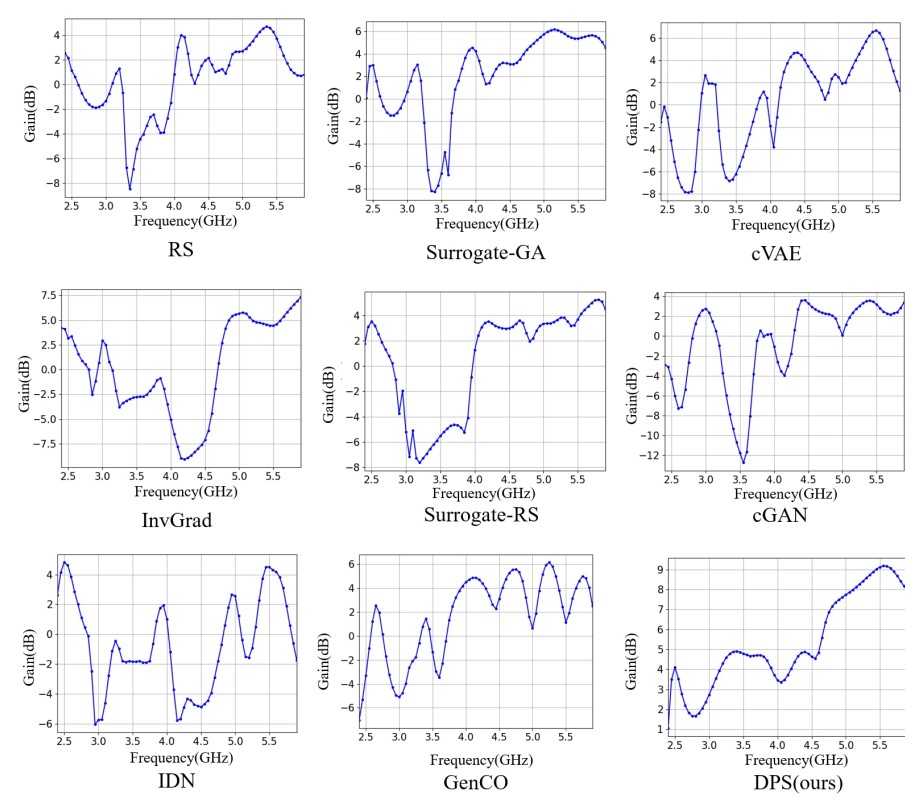

Figure 12: Simulated Results of Satisfactory Electromagnetic Structures on High-gain Antenna.

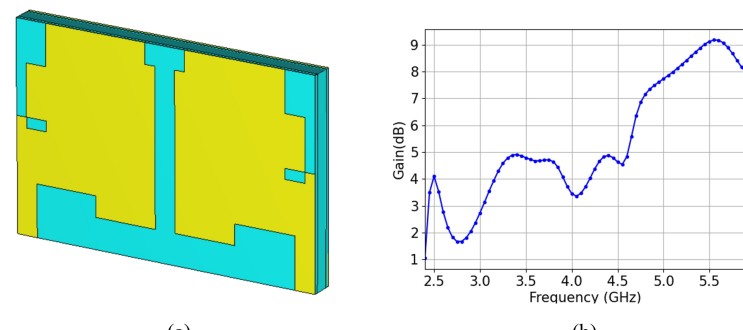

(a)            (b)

Figure 14: (a) Satisfactory Electromagnetic Structures of Different Methods on the High-gain Antenna. (b) Simulated Results of Satisfactory Electromagnetic Structures on High-gain Antenna.

