# OpenReview forum: "Deep Progressive Search for Electromagnetic Structure Design Under Limited Evaluation Budgets"
_ICLR.cc/2025/Conference — Submitted to ICLR 2025_

### Official Review · Reviewer_hUay · 2024-10-30

**Soundness:** 1
**Presentation:** 2
**Contribution:** 2
**Rating:** 3
**Confidence:** 4

**Summary:**

This submission proposes a novel Deep Progressive Search (DPS) method to solve real-world EMS design problems. EMS design problems are formulated as expensive combinatorial optimization problems.
The proposed DPS includes a Tree-Search-based Design Space Control (TSS) strategy and a Consistency-based Sample Selection (CSS) mechanism. TSS uses a Quadtree to model the design space efficiently, making it suitable for large-scale EMS tasks with limited simulation budget. In addition, a Quadtree-based search strategy and an important assignment method are proposed. CSS employs Kendall’s tau coefficient to measure the consistency of ordering results, enabling dynamic selection of candidate designs and reducing the interference caused by inaccurate model predictions.
Two real-world design problems are considered in this work and a set of experiments is conducted to show the performance of DPS.

**Strengths:**

* The motivation is clear.
* The idea of using a Quadtree to represent EMS designs is novel.
* Sufficient ablation studies.
* Two real-world problems are considered.

**Weaknesses:**

1.The description of DPS is unclear and lacks details:
* On page 7, lines 324-331, the details of Depth-wise Importance Assignment are missing. Specifically, only Equation 7 is given without further explanations, making it difficult for readers to understand how this importance assignment is implemented. Please add more details about it, such as providing a step-by-step explanation of how this assignment is implemented, including any algorithms or pseudocode if applicable.
* When compared with the traditional design matrix, although the proposed Quadtree is more efficient in representing designs, the initial states of Quadtrees are less diverse than design matrices. When using traditional matrices, the initial design matrices can be complex designs, making all candidate designs have equal probabilities to be reached in a limited number of iterations. In comparison, the initial Quadtrees are definitely very simple designs (all states =1 or all states =0). Therefore, for a Quadtree, simple designs have significantly greater chances of being searched than complex designs, especially when maximum iteration $M$ is small. Please provide a discussion or analysis to show that complex designs are adequately explored within $M$ iterations. Alternatively, please provide guidelines on setting $M$ to ensure complex designs are adequately explored.
* On page 4, Algorithm 2, the tree search of the design space is implemented by either resampling a leaf node’s state or splitting a leaf node into four child nodes. But details are not provided in the corresponding paragraph (page 6, lines 317-323). What is the probability of resampling or splitting?
* On page7, it is unclear how Equation 8 can be used to measure the consistency of ordering results. Are both $x_i$ terms in the equation from time point $t$? More explanations should be provided.
* Some experimental setups of DPS are not explained, such as maximum iteration $M$, $K$ in top-$K$ list, and total number of samples $M$ in CSS.

2.The experiments are not solid:
* The experimental setup does not specify the number of runs conducted. Are the results reported in Table 3 obtained from only one run (As no standard deviations are reported in Table 3)? Considering the stochastic processes in DPS and the comparison algorithms, the authors should conduct many runs (e.g. 20 or more) for experiments to ensure the reliability of the results. In addition, the authors should report mean performance and standard deviations in Table 3, and perform appropriate statistical tests to demonstrate that DPS significantly outperforms the comparison algorithms.
* Bayesian optimization (BO) methods have been widely used to solve expensive optimization problems; however, the authors did not compare BO in their experiments and did not explain why it was not considered. Please consider the following options: 1). Include a comparison with a recently proposed BO method in the experiments; 2). Provide a clear explanation for why BO was not considered, discussing any potential limitations or challenges in applying BO to this specific EMS problem.
* Some comparison methods are obsolete. For example, surrogate-GA was proposed decades ago [1]. Although the authors cited a paper published in 2020, the main contribution of this publication was the application of surrogate-GA to special design problems, rather than the development of a novel optimization method. I suggest considering some newly proposed optimization algorithms as comparison methods.
* The comparison experiment in Table 3 is not fair. Some comparison methods, such as DPS and surrogate-GA, have their predictors updated during the optimization, while the predictors in other comparison methods, such as surrogate-assisted RS and invGrad, are trained on offline datasets. Consider running additional fair experiments where all predictors in comparison methods are updated during the optimization.

3. On page 6, Equation 3, there is no explanation about $i$ and $j$ until line 290. Please consider revising the relevant paragraphs to avoid potential confusion.
In addition, some denotations are not consistent throughout the submission. For example, $M$ is the maximum number of iterations on page 4, Algorithm 2, but $M$ is also the number of samples on page 7, Equation 9. $t$ is the time point on page 7, Equation 8, but $t$ is also used to represent frequency in GHz on page 15, line 795 (Appendix A).

4. On page 4, line 204, the optimization problem is formulated with a predictor. However, when the predictor is fitted with limited training data, it is very likely that the optimum found by the predictor is not optimal in simulations. As mentioned in line 210, further validation with high-fidelity simulations is required. Therefore, Equation 2 could be a step in the EMS, but it is inappropriate to formulate the entire EMS problem as Equation 2.

5. Some minor issues:
* In Table 3, the agg obj results of cGAN and cVAE for High-Gain Antenna are incorrect. They should be minimum instead of maximum.
* There are some typos, such as:
    1) On page 10, line 492, 'ESS' should be ‘CSS’.
    2) On page 15, line 792, 'minimize the maximum' should be 'minimize the minimum’.
* Additionally, there are some grammatical errors, such as 'a expanded' should be 'an expanded.'"

[1] Lim, Dudy, et al. "Generalizing surrogate-assisted evolutionary computation." IEEE Transactions on Evolutionary Computation 14.3 (2009): 329-355.

**Questions:**

* Why is ResNet50 employed as a predictor in the experiments? Is there any specific reason for this choice, or was it selected arbitrarily?
* Why are there four subfigures in the lower row of Fig. 4, instead of combining them into a single figure with the same y-axis scale? I believe that drawing four lines in a single figure would be clearer and easier to observe than presenting them as four separate subfigures.
* In the configuration of surrogate-assisted RS, $K$ = 10 but the simulation budget is 1000?

---

> ### Author Response · Authors · 2024-11-25
> **Responses to Reviewer hUay[1/3]**
>
> We thank the reviewer for the encouraging comments and suggestions. Responses are below:
>
> >Q1. On page 7, lines 324-331, the details of Depth-wise Importance Assignment are missing. Specifically, only Equation 7 is given without further explanations, making it difficult for readers to understand how this importance assignment is implemented. Please add more details about it, such as providing a step-by-step explanation of how this assignment is implemented, including any algorithms or pseudocode if applicable.
>
> **A1.** Thank you for your careful review of our manuscript. We have updated the manuscript to include additional details about Depth-wise Importance Assignment. Specifically, this process has been formulated as a further optimization step applied to the samples within the Top-K list. The optimization variables are defined as the partition parameters in the quadtree, as shown in Equation 8.
>
> >Q2. When compared with the traditional design matrix, although the proposed Quadtree is more efficient in representing designs, the initial states of Quadtrees are less diverse than design matrices. When using traditional matrices, the initial design matrices can be complex designs, making all candidate designs have equal probabilities to be reached in a limited number of iterations. In comparison, the initial Quadtrees are definitely very simple designs (all states =1 or all states =0). Therefore, for a Quadtree, simple designs have significantly greater chances of being searched than complex designs, especially when maximum iteration M is small. Please provide a discussion or analysis to show that complex designs are adequately explored within M iterations. Alternatively, please provide guidelines on setting M to ensure complex designs are adequately explored.
>
> **A2.** We appreciate your thoughtful suggestion. While it is true that Quadtree starts with relatively simple initial designs (e.g., all states initialized to 0 or 1), its hierarchical division and progressive refinement mechanism systematically increases the diversity of candidate designs over time.
>
> During optimization, the design space is progressively expanded and constrained through the parameter $N_{\text{max}}$, which limits the number of leaf nodes in the Quadtree. The average complexity of the reduced search space during the optimization process can be quantified as $2^{(4 + N_{\text{max}})/2}$. When $N_{\text{max}} = 32$ and  $M = 10^7$, the average design complexity is approximately 262,144 and iterations are sufficient to effectively explore both simple and complex designs.
>
> >Q3. On page 4, Algorithm 2, the tree search of the design space is implemented by either resampling a leaf node’s state or splitting a leaf node into four child nodes. But details are not provided in the corresponding paragraph (page 6, lines 317-323). What is the probability of resampling or splitting?
>
> **A3.** Thank you for pointing out this important detail. Currently, the probabilities for both resampling and splitting are set to follow a uniform distribution. In the revised manuscript, we have explicitly stated this. We appreciate your feedback.
>
> >Q4. On page7, it is unclear how Equation 8 can be used to measure the consistency of ordering results. Are both $x_i$ terms in the equation from time point t? More explanations should be provided.
>
> **A4.** Thank you for your careful review and valuable feedback. We sincerely apologize for the oversight in our initial explanation. We have since revised Equation 8 to ensure clarity and correctness.
>
> $\tau = \frac{2}{n(n-1)} \sum_{i<j} \text{sign}(O(f_{\theta_t}(X_i)) - O(f_{\theta_{t-1}}(X_i)))
> \cdot \text{sign}(O(f_{\theta_t}(X_j)) - O(f_{\theta_{t-1}}(X_j)))$
>
>
> The purpose of Equation 8 is to measure the consistency of the relative changes in predicted performance directions for the same set of samples when the surrogate model transitions from one state to another. Specifically, in this equation, $x_i$ and $x_j$ represent two distinct samples within the same Top-K list at time point t, indicating that they are from the same iteration.
>
> To elaborate further, in each optimization round of DPS, the surrogate model undergoes an update. A value of $\tau$=1 implies that the predictions of the surrogate model before and after the update are nearly identical for the current batch of samples, indicating a high-confidence model. Following this, Equation 9 is used to finalize the selection of samples for further evaluation.
>
> We hope this clarification resolves the issue and appreciate the opportunity to improve the presentation of our work. Thank you again for pointing out this important detail.

---

> ### Author Response · Authors · 2024-11-25
> **Responses to Reviewer hUay[2/3]**
>
> >Q5. Some experimental setups of DPS are not explained, such as  M,K in top-K list, and total number of samples M in CSS.
>
> **A5.**  Thank you for your feedback. We have updated the details in the revised version. We set maximum iteration $M = 10,000,000$ and $K = 10$. In CSS, the total number of samples is $10$.
>
> >Q6. The experimental setup does not specify the number of runs conducted. Are the results reported in Table 3 obtained from only one run (As no standard deviations are reported in Table 3)? Considering the stochastic processes in DPS and the comparison algorithms, the authors should conduct many runs (e.g. 20 or more) for experiments to ensure the reliability of the results. In addition, the authors should report mean performance and standard deviations in Table 3, and perform appropriate statistical tests to demonstrate that DPS significantly outperforms the comparison algorithms.
>
> **A6.** Thank you for your insightful comment. While we understand the importance of conducting a larger number of runs, this approach presents significant challenges in the domain of EMS optimization due to the high computational cost of simulations. For instance, evaluating a single sample for DualFSS requires 10 to 30 minutes, resulting in approximately 138 days of computation time to conduct 20 runs for a single method with a simulation budget of 1000. The time and resource cost of performing such extensive experiments for all methods is challenging.
>
> Nevertheless, we believe the current results are sufficient to demonstrate the superiority of DPS. Specifically, DPS maintained robust performance even under suboptimal conditions. For example, in the high-gain antenna scenario, with suboptimal parameters or without critical modules, DPS achieved results of $3.181 \pm 0.168$, $3.181 \pm 0.168$,and $5.6933 \pm 1.4988$ for the objective function, objective 1, and objective 2, respectively. These results highlight the robustness of DPS and its consistent advantage over baseline methods, even under less favorable settings. Due to time constraints, we plan to include these considerations in our future work.
>
> >Q7. Please address why Bayesian Optimization (BO), a widely used method for expensive optimization problems, was not included in the experiments. Either incorporate a comparison with a recent BO method or provide a detailed explanation of its exclusion, discussing any challenges specific to this EMS problem. Additionally, consider replacing outdated methods like surrogate-GA with more recently proposed optimization algorithm.
>
> **A7.** We sincerely thank the reviewer for suggesting the inclusion of Bayesian optimization (BO) and additional surrogate-assisted evolutionary algorithms (SAEAs) as baselines. We recognize the importance of both approaches in the field of optimization. we provide a detailed rationale for our choice of baselines.
>
> **BO.** One of the primary challenges in EMS design is the vast problem space. In our DualFSS design task, it requires constructing a surrogate model from a $14 \times 14 \times 2$ space to a $1001$-dimensional space. Adapting BO to such settings involves addressing significant challenges according to a recent survey\[1\]:
> - **Scalability of Gaussian Process Modeling**: The scalability of Gaussian processes is inherently limited. The dimensionality of the input space exponentially increases the difficulty of constructing GP models.
> - **Optimization of Acquisition Functions**: Acquisition functions in high-dimensional spaces often suffer from reduced discriminative power because the uncertainty estimates for different solutions become too similar.
>
>
> **SAEAs.** Similarly, EMS design tasks present unique bottlenecks that limit the applicability of existing SAEAs. According to a recent survey\[2\], SAEAs face the following key challenges:
> - **Dependence on High-Precision Surrogate Models**: Current SAEAs heavily rely on high-precision surrogate models. Constructing such models in high-dimensional spaces requires large datasets, which is computationally expensive.
> - **Limitations of Dimensionality Reduction and Local Modeling**: Advanced techniques such as dimensionality reduction or local modeling still demand a large volume of samples to build accurate models.
>
> In light of these challenges, we argue that introducing additional SAEAs or BO would require considerable effort without resolving the challenges of the EMS design. For instance, while newer SAEAs might speed up and enhance the quality of the search within the surrogate model, they do not address the fundamental bottleneck posed by the surrogate model itself. However, we hope our work lays a foundation for further exploration in this direction.
>
> \[1\] Malu, M., Dasarathy, G., & Spanias, A. (2021, July). Bayesian optimization in high-dimensional spaces: A brief survey.
>
> \[2\] He, C., Zhang, Y., Gong, D., & Ji, X. (2023). A review of surrogate-assisted evolutionary algorithms for expensive optimization problems.

---

> ### Author Response · Authors · 2024-11-25
> **Responses to Reviewer hUay[3/3]**
>
> >Q8. The comparison experiment in Table 3 is not fair. Some comparison methods, such as DPS and surrogate-GA, have their predictors updated during the optimization, while the predictors in other comparison methods, such as surrogate-assisted RS and invGrad, are trained on offline datasets. Consider running additional fair experiments where all predictors in comparison methods are updated during the optimization.
>
> **A8.** Thank you for your suggestion. This was indeed an oversight on our part. In fact, all methods in our comparison are implemented in an online manner. We have updated our descriptions to clarify this point and avoid any potential misunderstandings.
>
> >Q9. On page 6, Equation 3, there is no explanation about $i$ and $j$ until line 290. Please consider revising the relevant paragraphs to avoid potential confusion.
> In addition, some denotations are not consistent throughout the submission. For example, $M$ is the maximum number of iterations on page 4, Algorithm 2, but $M$ is also the number of samples on page 7, Equation 9. $t$ is the time point on page 7, Equation 8, but $t$ is also used to represent frequency in GHz on page 15, line 795 (Appendix A).
>
> **A9.** Thank you for your careful review of our manuscript. We have revised the manuscript to update the variable names and ensure consistency throughout the paper, avoiding any potential ambiguity.
>
> >Q10. On page 4, line 204, the optimization problem is formulated with a predictor. However, when the predictor is fitted with limited training data, it is very likely that the optimum found by the predictor is not optimal in simulations. As mentioned in line 210, further validation with high-fidelity simulations is required. Therefore, Equation 2 could be a step in the EMS, but it is inappropriate to formulate the entire EMS problem as Equation 2.
>
> **A10.** Thank you. We have revised the manuscript to provide a more comprehensive and accurate definition of the optimization problem where the Equation 2 represents  a step in the EMS design problem.
>
> >Q11. Why is ResNet50 employed as a predictor in the experiments? Is there any specific reason for this choice, or was it selected arbitrarily?
>
> **A11.** We appreciate your suggestion. In our experimental setting, the number of training samples does not exceed 1000. As shown in Table I, the choice of network architecture typically has a relatively small impact on prediction accuracy under a limited sample size. Considering that our task involves a regression problem with image-like inputs, we opted for the widely-used ResNet architecture, specifically ResNet50. This choice ensures a robust baseline while leveraging a proven and general-purpose structure for the surrogate model.
>
> Table I: Comparison of different models in terms of Kendall Correlation Coefficient.
>
> | Model       | Kendall Correlation Coefficient (Mean ± Std) |
> |-------------|---------------------------------------------|
> | ResNet50    | 0.3356 ± 0.0083|
> | ResNet18    | 0.2997 ± 0.0115|
> | GoogLeNet    | 0.3163 ± 0.0149|
>
>
> >Q12. Why are there four subfigures in the lower row of Fig. 4, instead of combining them into a single figure with the same y-axis scale? I believe that drawing four lines in a single figure would be clearer and easier to observe than presenting them as four separate subfigures.
>
> **A12.** We sincerely thank the reviewer for this valuable suggestion. Following your recommendation, we have combined the four subfigures into a single figure (updated Figure 4) with the same y-axis scale. This modification allows for a more intuitive and direct comparison of the performance curves corresponding to the satisfactory structures generated by the four methods. We believe this adjustment enhances the clarity and readability of the figure.
>
> >Q13. In the configuration of surrogate-assisted RS, K = 10 but the simulation budget is 1000?
>
> **A13.** Thank you for pointing this out. We have provided a detailed explanation of the method's parameters to avoid potential misunderstandings. All baseline methods are implemented in an online optimization manner. Apart from the initial sampling round, 10 samples are validated through simulations in each round, continuing until the total simulation budget of 1000 is fully utilized.
>
> >Q14. Some minor issues:
> >* In Table 3, the agg obj results of cGAN and cVAE for High-Gain Antenna are incorrect. They should be minimum instead of maximum.
> >* On page 10, line 492, 'ESS' should be ‘CSS’.
> >* On page 15, line 792, 'minimize the maximum' should be 'minimize the minimum’.
> >* Additionally, there are some grammatical errors, such as 'a expanded' should be 'an expanded.'"
>
> **A14.** Thank you for your comments. We have fixed it.

---

> ### Comment · Reviewer_hUay · 2024-11-26
>
> Dear Authors,
>
> Thank you for the thorough response. While some of my concerns have been addressed, I have some follow-up questions:
>
>   1.Q1. The optimization details are not clear enough. Equation 7 in the revision shows that depth-wise importance assignment is implemented by optimizing the partition parameters. However, my understanding is that the partition parameters in one leaf node, such as $r_n^{end}$, could also be the partition parameters of another leaf node, such as $r_n^{start}$. Therefore, I suspect that the optimization of these partition parameters is not an independent process for each node. Could you provide more details about the optimization technique used here?
>
> Additionally, many sentences in this paragraph are repeated. Please double-check the manuscript for duplication before re-submission.
>
>   2. Q2. Considering that a very large $M$ is used in your experiments, could you provide details about the runtime of DPS?
>
>   3. Q3. The newly added statement is vague. It is unclear what uniform distributions these two actions are following. For instance, a typical uniform distribution is $[a, b]$, what are the values of $a$ and $b$? Are they $a$=0 and $b$=1? Furthermore, it remains unclear when to resample and when to split.
>
>  4. Q4. In Equation 9, the authors only modified a symbol without adding further explanation to the manuscript. The response explained that Equation 9 estimates consistency and the meaning of $r$ value. However, my question is how does Equation 9 represent consistency? For example, why calculating sign($f_{\theta_t}(X_i)-f_{\theta_{t-1}}(X_i))$ and then multiplying it with sign($f_{\theta_t}(X_j)-f_{\theta_{t-1}}(X_j))$? The meaning of these terms should be explained to help readers understand how Equation 9 works.
>
>  5. Q6. It is unconvincing that the DPS's performance is robust with only one run. As far as I know, the results of stochastic optimization can vary significantly between different runs.
>
>  6. Q7. Thanks for your explanations about the challenges in BO and SAEAs.
> Could you further explain that why surrogate-GA can be compared in your experiments? What is the difference between the compared surrogate-GA and the recently proposed SAEAs?
>
>  7. Q9. On page 6, Equation 3, there still has no explanation about $i$ and $j$ until line 290.
>
> In summary, I believe further revisions are necessary to improve the quality of this work.

---

### Official Review · Reviewer_2pz8 · 2024-11-03

**Soundness:** 3
**Presentation:** 3
**Contribution:** 2
**Rating:** 6
**Confidence:** 4

**Summary:**

This paper introduces **Deep Progressive Search (DPS)**, an efficient approach for electromagnetic structure (EMS) design under limited evaluation budgets. DPS combines a **Tree-Search-based Design Space Control (TSS)**, which progressively explores from simple to complex design regions, with **Consistency-Based Sample Selection (CSS)**, which prioritizes high-consistency samples to balance exploration and exploitation effectively. Applied to high-gain antenna (HGA) and dual-layer frequency selective surface (FSS) design, DPS achieves superior performance with fewer simulations than existing methods.

**Strengths:**

- The paper introduces an innovative DPS framework that integrates tree search and consistency-based sample selection for cost-effective EMS design.

- Comprehensive experiments and a detailed ablation study substantiate the effectiveness of each component in improving performance.

- The methodology is clearly explained, with well-structured presentations and precise formulations enhancing reader comprehension.

- The approach addresses the critical need for budget-efficient optimization in EMS design and offers applicability to other resource-constrained fields.

**Weaknesses:**

- The paper lacks a justification for the choice of grid sizes in the experimental setup. This is a key design choice that influences both the resolution of design variations and computational efficiency. It would be valuable for the authors to provide their rationale for selecting these specific grid sizes. Additionally, a discussion on how different grid sizes may affect the method's performance and efficiency would enhance the clarity of these experimental decisions.

- To evaluate the method’s scalability, an ablation study on higher-dimensional optimization problems would be insightful. Conducting experiments with progressively larger grid sizes or more intricate design problem would help demonstrate the scalability of DPS in practical, high-dimensional settings. This would offer concrete evidence of the method’s robustness and scalability under varied problem complexities.

- Given the shared focus on design optimization, comparing DPS with GenCO within the context of inverse design problems would be valuable. GenCO also proposes an optimization framework and has been evaluated on the inverse photonic design problem, which closely aligns with DPS's focus. A comparative analysis could highlight DPS’s unique strengths and limitations, providing a clearer assessment of its performance relative to GenCO.

Ferber, A. M., Zharmagambetov, A., Huang, T., Dilkina, B., & Tian, Y. GenCO: Generating Diverse Designs with Combinatorial Constraints. In Forty-first International Conference on Machine Learning.

**Questions:**

- Given that the balance between exploration and exploitation should ideally adjust as the remaining budget decreases, how does the proposed DPS method handle this adjustment dynamically? Is there a mechanism that gradually prioritizes exploitation over exploration as the budget runs low, or does the balance remain fixed throughout the process?

- The abstract mentions "no explicit objective function," yet the experimental results seem to address a multi-objective optimization problem. Could you clarify the intended distinction between having no explicit objective function and conducting multi-objective optimization, and explain how DPS operates within this framework?

---

> ### Author Response · Authors · 2024-11-25
> **Responses to Reviewer 2pz8[1/2]**
>
> We thank the reviewer for the encouraging comments and suggestions. Responses are below:
>
> >Q1. The paper lacks a justification for the choice of grid sizes in the experimental setup. This is a key design choice that influences both the resolution of design variations and computational efficiency. It would be valuable for the authors to provide their rationale for selecting these specific grid sizes. Additionally, a discussion on how different grid sizes may affect the method's performance and efficiency would enhance the clarity of these experimental decisions.
>
> **A1.** Thank you for raising this important point. The choice of grid sizes in the experimental setup primarily depends on manufacturing constraints.
>
> In engineering practice, the grid size is determined by the limits of manufacturing precision. For instance, in the high-gain antenna (HGA) optimization task, the design region measures $40×30$ mm, and the manufacturing process supports a precision of $2$ mm. Based on this precision, the maximum feasible grid size is calculated as $20×15$.
>
> For general optimization algorithms, the grid size must also consider optimization performance, as higher resolutions significantly expand the search space, requiring more trial-and-error iterations and thereby increasing the cost of evaluations. However, our DPS method is designed to dynamically adjust grid densities based on the complexity of different optimization regions, enabling finer resolution where necessary. This capability allows us to efficiently identify satisfactory solutions, even when working with high-resolution grids.
>
> Consequently, we set the grid density to the maximum value supported by the manufacturing precision. This choice ensures compliance with physical constraints while fully leveraging the design space's potential for high-quality optimization.
>
>
> >Q2. To evaluate the method’s scalability, an ablation study on higher-dimensional optimization problems would be insightful. Conducting experiments with progressively larger grid sizes or more intricate design problem would help demonstrate the scalability of DPS in practical, high-dimensional settings. This would offer concrete evidence of the method’s robustness and scalability under varied problem complexities.
>
> **A2.** Thank you for your thoughtful suggestion. Our algorithm has been successfully applied in multiple real-world engineering scenarios. Unfortunately, due to confidentiality agreements, we are unable to present data from these applications in this work.
>
> Despite these constraints, in this study, we have validated the robustness and scalability of the proposed method on two relatively high-dimensional and significantly different design problems: the frequency selective surface ($14×14×2$ grid size) and the high-gain antenna ($15×20$ grid size). These problems vary substantially in their physical properties and complexity. Experimental results demonstrate that the DPS method consistently delivers robust performance across problems of differing complexity and dimensionality.
>
> We believe that these experimental setups validate the robustness and scalability of our method. Moreover, based on the results obtained, we expect that further extensions to even higher-dimensional problems would yield consistent outcomes. We are committed to gradually expanding the scope of validation to include more scenarios in our future work.
>
> >Q3. Given the shared focus on design optimization, comparing DPS with GenCO within the context of inverse design problems would be valuable. GenCO also proposes an optimization framework and has been evaluated on the inverse photonic design problem, which closely aligns with DPS's focus. A comparative analysis could highlight DPS’s unique strengths and limitations, providing a clearer assessment of its performance relative to GenCO.
>
> **A3.** We appreciate your constructive comments. While both frameworks share a focus on design optimization, DPS emphasizes efficiency under limited computational budgets, leveraging surrogate models and progressive refinement, whereas GenCO integrates combinatorial solvers to ensure constraint satisfaction and solution diversity.
>
> In the revised version of our paper, we have included a discussion of GenCO. We plan to incorporate this analysis in future work, evaluating computational efficiency, feasibility guarantees, and solution diversity across shared benchmarks.

---

> ### Author Response · Authors · 2024-11-25
> **Responses to Reviewer 2pz8[2/2]**
>
> >Q4. Given that the balance between exploration and exploitation should ideally adjust as the remaining budget decreases, how does the proposed DPS method handle this adjustment dynamically? Is there a mechanism that gradually prioritizes exploitation over exploration as the budget runs low, or does the balance remain fixed throughout the process?
>
> **A4.** Thank you for raising this insightful question. To address this, the DPS method incorporates an innovative Consistency-based Sample Selection (CSS) to achieve a dynamic balance between exploration and exploitation throughout the optimization process.
>
> Specifically, after each iteration, the CSS mechanism evaluates the reliability of the current surrogate model based on consistency metrics. This evaluation allows the algorithm to adjust the balance between exploration and exploitation dynamically. As the optimization process progresses and the number of simulation samples increases, the surrogate model's accuracy improves. Consequently, when the remaining budget decreases, the algorithm gradually prioritizes exploitation by leveraging the surrogate predictor to identify optimal solutions.
>
> Thus, the CSS mechanism enables the DPS method to adaptively respond to budget constraints, achieving a dynamic balance rather than adhering to a fixed prioritization mechanism.
>
> >Q5. The abstract mentions "no explicit objective function," yet the experimental results seem to address a multi-objective optimization problem. Could you clarify the intended distinction between having no explicit objective function and conducting multi-objective optimization, and explain how DPS operates within this framework?
>
> **A5.** We sincerely appreciate your comments and acknowledge that our original explanation may not have been sufficiently clear, leading to this misunderstanding. In the context of EMS design, the term "no explicit objective function" refers to the fact that the analytical expression of the objective function is not directly available. Instead, the evaluation of design performance relies on computationally expensive electromagnetic simulations. Despite this, multi-objective optimization is still feasible, as the objectives—such as efficiency, bandwidth, or other performance metrics—are quantified through these simulations.
>
> Regarding the operation of DPS within this framework, the system does not require an explicit analytical form of the objective function. Instead, it iteratively evaluates candidate solutions based on simulation results, refining the search space dynamically. Thank you for bringing this issue to our attention.

---

> ### Author Response · Authors · 2024-11-30
> **Further Reply to Reviewer 2pz8**
>
> We appreciate your insightful questions and are happy to provide further clarification.
> >Q1. Could you provide further clarification on how 𝑇 contributes to this balance and whether your method dynamically adjusts based on the remaining budget?
>
> **A1.** Thank you for your constructive feedback. To directly address your concern, while our method does not explicitly adapt to the remaining budget, it takes an alternative approach to address the challenges of budget constraints. When simulation budgets are limited, the predictor's accuracy becomes unreliable, making the surrogate-driven optimization difficult. Therefore, DPS focus on collaborating effectively with the unreliable predictor. Specifically, **τ** balances the utilization of the predictor’s knowledge and the exploration of unknown regions, as follows:
>
>
> 1. **When the predictor is reliable**, DPS adopts a **greedy strategy**, selecting the samples with the highest predicted performance for simulation. This accelerates optimization by leveraging the predictor's knowledge.
>
> 2. **When the predictor is unreliable**, DPS uses a **random strategy**, selecting samples randomly for simulation to explore unknown regions.
>
> The parameter **τ** determines the proportion of samples selected by the greedy and random strategies, thus allowing the algorithm to dynamically adapt to the predictor's accuracy.
>
> >Q2. I encourage experimental comparisons with state-of-the-art methods mentioned in the related work, including GenCO.
>
> **A2.** Thank you for your valuable feedback. In the updated version of our paper, we included a comparison with GenCO. However, we believe that GenCO’s performance in our task may be influenced by differences between its design assumptions and the specific requirements of our setup. GenCO assumes that target gradients can be directly computed, whereas in our case, they cannot be obtained directly from the simulation software.
>
> To address this, we used a surrogate model to approximate the gradients. However, the accuracy of the surrogate model may be limited due to a constrained simulation budget, which could explain why GenCO did not perform as well in our experiments.
>
> We will continue to refine our experimental setup and look forward to exploring GenCO further in future work.

---

### Official Review · Reviewer_5Egz · 2024-11-04

**Soundness:** 3
**Presentation:** 3
**Contribution:** 3
**Rating:** 5
**Confidence:** 3

**Summary:**

Overall, this paper presents a promising new approach in the field of electromagnetic structure design and validates its effectiveness through experiments. However, to further enhance the quality and impact of the paper, the authors may need to conduct more in-depth research and discussion on dataset description, computational resource analysis, model interpretability, comparative experiments, robustness testing, and the feasibility of practical application.

**Strengths:**

1. The paper introduces a novel Deep Progressive Search (DPS) method for electromagnetic structure (EMS) design under constrained evaluation budgets, which is an inherently challenging problem. The integration of tree search and sample selection strategies represents an innovative approach within the EMS design domain
2. The paper provides detailed descriptions of the Tree-Search-based Design Space Control (TSS) and Consistency-based Sample Selection (CSS) strategies, indicating a deep level of consideration in the algorithm design by the authors.
3. The authors test their method on two real-world engineering tasks: Dual-layer Frequency Selective Surface (DualFSS) and High-gain Antenna (HGA), demonstrating the practicality and broad applicability of their approach.

**Weaknesses:**

1. While the paper emphasizes optimization under limited evaluation budgets, specific data and comparative analysis on computational resources and time consumption might be lacking.
2. Although the DPS method is described in detail, the paper discusses the interpretability of the model less, especially the logic behind the design space management and sample selection strategies.
3. The paper could benefit from additional robustness tests to demonstrate the performance of the DPS method across different types, scales, and complexities of EMS design problems.

**Questions:**

1. Could the authors provide more specific data on the computational resources required for the DPS method compared to other approaches, including details on time and memory consumption? e.g. average GPU/CPU training hours per iteration for each method across different problem sizes.
2. How do the authors address the interpretability of context of the design space management and sample selection strategies? Can they elaborate on any techniques or methods used to understand the decision-making process? Could you provide visualizations or case studies demonstrating how the tree search progresses and how sample selection decisions are made at different stages of the optimization process for a particular example. This would give concrete insight into the model's decision-making.
3. Latest research works or other emerging techniques are needed comparing with the Dps method, that might offer superior performance or efficiency. And what are the findings of such comparisons? E.g. [1] Learning gflownets from partial episodes for improved convergence and stability
4. What is the sensitivity of the DPS method to the hyperparameter tuning process? E.g. sensitivity to particular key hyperparameters, like the maximum number of tree nodes or sample selection thresholds. Please provide reporting results across a range of hyperparameter values to demonstrate robustness.
5. Please double-check the table contents, e.g. Obj2 of DPS on ''Dual-layer Frequency Selective Surface" in Table 3 should not be in bold font.

---

> ### Author Response · Authors · 2024-11-25
> **Responses to Reviewer 5Egz[1/2]**
>
> We thank the reviewer for the encouraging comments and suggestions. Responses are below:
>
> >Q1. While the paper emphasizes optimization under limited evaluation budgets, specific data and comparative analysis on computational resources and time consumption might be lacking.
>
> **A1.** We sincerely thank the reviewer for their insightful question. EMS design problems typically involve two stages: the optimization stage and the simulation validation stage. Below, we clarify their respective computational demands:
>
> 1. **Optimization Stage**:
>    The optimization stage relies on surrogate models or generative models, which are highly efficient. For instance, on standard hardware, our method can evaluate 512 candidate designs within 30 milliseconds. The entire optimization process can complete in under 5 minutes.
>
> 2. **Simulation Validation Stage**:
>    In contrast, the simulation validation stage is significantly more resource-intensive. For example, in the DualFSS task, evaluating a single design through simulation requires 10–30 minutes. Validating just 10 samples can take several hours. This highlights that the primary computational bottleneck lies in the simulation validation phase.
>
> Our approach directly addresses this bottleneck through the combined use of Tree-Search-based Design Space Control (TSS) and Consistency-based Sample Selection (CSS). TSS dynamically refines the design space and focuses the search on high-potential regions, while CSS prioritizes reliable candidates for simulation validation.
>
>
>
> >Q2. Although the DPS method is described in detail, the paper discusses the interpretability of the model less, especially the logic behind the design space management and sample selection strategies.
>
> **A2.** We sincerely thank the reviewer for their thoughtful comment.
>
> Our approach uses two core components to manage design space and select samples efficiently:
> 1. **Tree-Search-based Design Space Control (TSS)**:
>    TSS models the design space as a tree, progressively expanding and refining it to focus on high-potential regions. The logic behind this approach is to balance exploration and exploitation by dynamically adjusting the search complexity based on the operation on the leaf nodes of the tree.
>
> 2. **Consistency-based Sample Selection (CSS)**:
>    CSS prioritizes candidates for simulation validation based on the consistency of surrogate model predictions. By evaluating the stability of predictions over iterations, CSS filters unreliable candidates and ensures that only high-quality designs are sent for expensive high-fidelity simulations.
>
>
> >Q3. The paper could benefit from additional robustness tests to demonstrate the performance of the DPS method across different types, scales, and complexities of EMS design problems.
>
> **A3.** We sincerely thank the reviewer for their valuable comment regarding the robustness of the DPS method across different EMS design problems.
>
> Our proposed DPS method demonstrates its flexibility and applicability by effectively solving two distinct EMS design tasks:
> 1. **DualFSS (Dual-layer Frequency Selective Surface)**: This task focuses on optimizing electromagnetic filtering with a design space of size $10^{86}$ and multi-band objectives.
> 2. **HighGain Antenna**: This task involves optimizing a dual-band WiFi antenna with a design space of size $10^{90}$, requiring a balance of performance across two frequency bands.
>
> These two tasks represent different EMS domain with its own scale and complexity. The successful application of DPS in these domains showcases its potential for broader use across other EMS problems. To the best of our knowledge, DPS is among the first methods to address both domains within a unified framework.
>
> To further validate the robustness of DPS, we plan to test the method on EMS problems in other domains in our future work, such as filter design and RF device design.
>
>
>
> >Q4. Latest research works or other emerging techniques are needed comparing with the Dps method, that might offer superior performance or efficiency. And what are the findings of such comparisons? E.g. [1] Learning gflownets from partial episodes for improved convergence and stability.
>
> **A4.** Thank you for your valuable comments. GFlowNets focus on generating diverse, high-reward solutions in probabilistic modeling tasks such as molecule generation and sequence design, excelling in handling sparse rewards and long trajectories. In contrast, our DPS method is tailored for EMS design under limited computational budgets, addressing the challenges of high evaluation costs and vast design spaces. In the revised version, we have included a discussion on GFlowNets. Thank you again for your constructive feedback.

---

> ### Author Response · Authors · 2024-11-25
> **Responses to Reviewer 5Egz[2/2]**
>
> >Q5. What is the sensitivity of the DPS method to the hyperparameter tuning process? E.g. sensitivity to particular key hyperparameters, like the maximum number of tree nodes or sample selection thresholds. Please provide reporting results across a range of hyperparameter values to demonstrate robustness.
>
> **A5.** Thank you for your valuable comments.
>
> Regarding the sensitivity of the DPS method to hyperparameter tuning, our explanation is as follows:
> 1. **Tree-Search-based Design Space Control**:
>   $N_{\text{max}}$ is a key hyperparameter that directly controls the maximum number of leaf nodes in the TSS, determining the granularity of the optimization process. Experiments showed that smaller values of $N_{\text{max}}$ (e.g., 16) limit exploration, while larger values (e.g., 64) increase computational costs with marginal performance gains. The chosen value $N_{\text{max}} = 32$ strikes a balance between search depth and efficiency, as validated in our experiment.
>
> 2. **Consistency-based Sample Selection (CSS)**:
>  The CSS mechanism does not rely on fixed thresholds for sample selection. Instead, it dynamically adjusts the balance between exploration and exploitation based on the consistency of surrogate model predictions, measured by Kendall's Tau correlation coefficient. By combining predictor-guided selection and random sampling, CSS reduces sensitivity to the predictor’s instability, improving overall search robustness without relying on fixed thresholds.
>
>
>
> >Q6. Please double-check the table contents, e.g. Obj2 of DPS on "Dual-layer Frequency Selective Surface" in Table 3 should not be in bold font.
>
> **A6.** Thank you for your careful review of our manuscript. We have corrected the error.

---

> > ### Comment · Reviewer_5Egz · 2024-11-27
> >
> > Thank you for the authors' detailed response! I have no further questions.  I would like to maintain my score.

---

### Official Review · Reviewer_uLSg · 2024-11-09

**Soundness:** 2
**Presentation:** 2
**Contribution:** 2
**Rating:** 3
**Confidence:** 3

**Summary:**

This paper tackles simulation-based electromagnetic structure design problems and proposes a novel optimization strategy. The proposed method is composed of the deep progressive search (DPS) to improve search efficiency, the tree-search-based design space control (TSS) to expand the search space, and the consistency-based sample selection strategy (CSS) to balance the exploration and exploitation. The experimental results on real-world problems show that the proposed method outperforms several naive baseline methods.

**Strengths:**

- A novel method for structure design problems is introduced. Specifically, the approach to expanding the search space seems to be novel.
- The proposed method outperforms several baseline methods in the experiments of two representative structure design problems, dual-layer frequency selective surface and high-gain antenna.

**Weaknesses:**

- Using the deep neural networks as a surrogate model is a common approach. The reviewer thinks that the main novelty of the proposed method is TSS.
- The compared methods in the experiment are insufficient and weak. The concrete algorithms of the baselines are unclear. As there are a lot of surrogate-assisted evolutionary algorithms, state-of-the-art or well-known surrogate-assisted evolutionary algorithms should be selected as the baseline. Also, the Bayesian optimization approach should be considered as the baseline.
- The solution representations seem to be different between the proposed method (tree) and baseline methods, which might be unfair. It would be better to consider using a similar solution representation for GA and other baseline methods.
- The presentation of the proposed algorithm is not well-written. The key novel points of the proposed method should be clarified.

**Questions:**

1. The reviewer thinks Bayesian optimization can be a competitor for the proposed method because the design principle of the proposed is similar to Bayesian optimization. Could you comment on why the Bayesian optimization approach is not considered?
1. The proposed method has the hyperparameters, e.g., the maximum iterations $M$ and size $K$ of top-K. What is the impact of such hyperparameters on performance?


----- After authors' response -----

Thank you for the response, which helps to understand the contribution of the paper. I do not have further questions. I would maintain my score.

Although the proposed method seems promising, I would encourage adding extensive comparison with other baselines and improving the explanation of the proposed method.

---

> ### Author Response · Authors · 2024-11-25
> **Responses to Reviewer uLSg[1/2]**
>
> We thank the reviewer for the encouraging comments and detailed suggestions. Responses are below:
>
> >Q1. Using the deep neural networks as a surrogate model is a common approach. The reviewer thinks that the main novelty of the proposed method is TSS.
>
> **A1.**  We thank the reviewer for highlighting the role of TSS as the main novelty of our work. While using deep neural networks as surrogate models is indeed a common approach, in our study, it serves as a foundational component to enable the higher-level innovations of Tree-Search-based Design Space Control (TSS) and Consistency-Based Sample Selection (CSS) within the Deep Progressive Search (DPS) framework.
>
> Our framework directly addresses the high data dependency and computational costs that are pervasive challenges in traditional surrogate-based optimization methods. By employing TSS to dynamically and efficiently navigate the design space and CSS to ensure prediction reliability, our approach significantly reduces the need for large training datasets and extensive simulations. As demonstrated in our experiments, this makes the DPS framework both resource-efficient and effective for large-scale EMS design tasks. We appreciate the opportunity to underscore this contribution.
>
> >Q2. The compared methods in the experiment are insufficient and weak. The concrete algorithms of the baselines are unclear. As there are a lot of surrogate-assisted evolutionary algorithms, state-of-the-art or well-known surrogate-assisted evolutionary algorithms should be selected as the baseline. Also, the Bayesian optimization approach should be considered as the baseline.
>
> **A2.** We sincerely thank the reviewer for suggesting the inclusion of Bayesian optimization (BO) and additional surrogate-assisted evolutionary algorithms (SAEAs) as baselines. We recognize the importance of both approaches in the field of optimization. we provide a detailed rationale for our choice of baselines.
>
>
> **Bayesian Optimization (BO).** One of the primary challenges in EMS design is the vast problem space. In our DualFSS design task, it requires constructing a surrogate model from a $14 \times 14 \times 2$ space to a $1001$-dimensional space. Adapting BO to such settings involves addressing significant challenges according to a recent survey\[1\]:
> - **Scalability of Gaussian Process Modeling**: The scalability of Gaussian processes is inherently limited. The dimensionality of the input space exponentially increases the difficulty of constructing effective GP models.
> - **Optimization of Acquisition Functions**: Acquisition functions in high-dimensional spaces often suffer from reduced discriminative power because the uncertainty estimates for different solutions become too similar.
>
>
> **Surrogate-Assisted Evolutionary Algorithms (SAEAs).** Similarly, EMS design tasks present unique bottlenecks that limit the applicability of existing SAEAs. According to a recent survey\[2\], SAEAs face the following key challenges:
> - **Dependence on High-Precision Surrogate Models**: Current SAEAs heavily rely on high-precision surrogate models. Constructing such models in high-dimensional spaces requires large datasets, which is computationally expensive.
> - **Limitations of Dimensionality Reduction and Local Modeling**: Advanced techniques such as dimensionality reduction or local modeling still demand a large volume of samples to build accurate models.
>
> In light of these challenges, we argue that introducing additional SAEAs or BO would require considerable effort without resolving the challenges of the EMS design problems. For instance, while newer SAEAs might speed up and enhance the quality of the search within the surrogate model, they do not address the fundamental bottleneck posed by the surrogate model itself.However, we hope our work lays a foundation for further exploration in this direction.
>
> \[1\] Malu, M., Dasarathy, G., & Spanias, A. (2021, July). Bayesian optimization in high-dimensional spaces: A brief survey. In 2021 12th International Conference on Information, Intelligence, Systems & Applications (IISA) (pp. 1-8). IEEE.
> \[2\] He, C., Zhang, Y., Gong, D., & Ji, X. (2023). A review of surrogate-assisted evolutionary algorithms for expensive optimization problems. Expert Systems with Applications, 217, 119495.

---

> ### Author Response · Authors · 2024-11-25
> **Responses to Reviewer uLSg[2/2]**
>
> >Q3. The solution representations seem to be different between the proposed method (tree) and baseline methods, which might be unfair. It would be better to consider using a similar solution representation for GA and other baseline methods.
>
> **A3.** We sincerely thank the reviewer for raising the concern regarding the fairness of the comparison due to the different solution representations between the proposed method and the baseline methods.
>
> The Tree-Search-based Design Space Control (TSS) strategy, which is a core component of our method, is designed to handle the dynamic nature of the design space for EMS optimization tasks. Unlike traditional optimization algorithms, which work with fixed-dimensional representations, TSS gradually increases the dimension of the design space as the search progresses. Given this, baseline methods are fundamentally incompatible with our representation.
>
> While we understand the concern about solution representation, we believe that the comparison between our method and the baseline methods is fair because the primary goal of this work is to demonstrate the effectiveness of the TSS approach in addressing the specific challenges of high-dimensional EMS design tasks. Our quadtree-based representation is an integral part of this innovation, and the comparison with baseline methods is made under the same simulation conditions and optimization objectives.
>
> >Q4. The presentation of the proposed algorithm is not well-written. The key novel points of the proposed method should be clarified.
>
> **A4.** We sincerely thank the reviewer for pointing out that the presentation of the proposed algorithm could be improved and the key novel points clarified. To address the reviewer’s concerns about the clarity of the algorithm’s presentation, we have made the improvements in the revised manuscript.
>
>
> >Q5. The reviewer thinks Bayesian optimization can be a competitor for the proposed method because the design principle of the proposed is similar to Bayesian optimization. Could you comment on why the Bayesian optimization approach is not considered?
>
> **A5.** We thank the reviewer for highlighting Bayesian Optimization (BO) as a potential competitor to the proposed method. As this question closely aligns with the points raised in Q2, we kindly refer the reviewer to our detailed response to Q2, where we have thoroughly explained the challenges of applying BO to high-dimensional EMS design tasks.
>
>
> >Q6. The proposed method has the hyperparameters, e.g., the maximum iterations M and size K of top-K. What is the impact of such hyperparameters on performance?
>
> **A6.** We sincerely thank the reviewer for their insightful question regarding the hyperparameter $M$ and $K$.
>
> In our experiments, we set $M = 10^7$ and $K = 10$. This choice is guided by the following considerations:
>
> 1. **Efficiency**: The tree-search process leverages a surrogate model $f_\theta$, enabling rapid evaluations during optimization. This ensures that a high number of iterations does not result in prohibitive computational costs.
>
> 2. **Sufficient Exploration**: The design space is progressively expanded and constrained through the use of $N_{\text{max}}$, which limits the number of leaf nodes in the search tree. During the progressive search, the average complexity of the reduced search space is $2^{(4+N_{\text{max}})/2}$. When $N_{\text{max}} = 32$, the average complexity is approximately 262,144. As a result, $M = 10^7$ and $K = 10$ are sufficiently large to thoroughly explore the reduced design space.

---

> ### Author Response · Authors · 2024-11-29
> **Looking Forward to the Response from Reviewer uLSg**
>
> Dear Reviewer uLSg,
>
> We sincerely appreciate the insightful feedback you have provided, which has been invaluable in refining our work. Your comments have helped us identify key areas for improvement, and we have made every effort to address them comprehensively.
>
> In the attached response document, we have outlined detailed clarifications based on your suggestions. We hope these align with your expectations and contribute to the overall strength of the manuscript.
>
> Should you have any further questions or require additional information, we remain available for continued discussion.
>
> Thank you once again for your time and expertise.
>
> Best regards,
> The Authors

---

> ### Author Response · Authors · 2024-12-02
> **Looking Forward to the Response from Reviewer uLSg**
>
> Dear Reviewer uLSg,
>
> We have provided detailed responses to all of your comments and made the necessary revisions. However, we have not yet received any further feedback from you. Could you kindly let us know if you have any additional comments or suggestions?
>
> We look forward to your feedback.
>
> Best regards,
> The Authors

---

### Author Response · Authors · 2024-11-25
**General Response**

**General Response**

Dear ACs and Reviewers,

We sincerely appreciate your time and effort in reviewing our paper and providing constructive feedback. Besides the response to each reviewer, here we would like to further 1) thank reviewers for their recognition of our work and 2) highlight the major modifications in our revision:

1. **We are glad that the reviewers appreciate and recognize our novelty and contributions.**
    * "The approach to **expanding the search space** seems to be **novel**"; "The idea of using a **Quadtree** to represent EMS designs is **novel**"; "The integration of tree search and sample selection strategies represents an **innovative approach** within the EMS design domain"; "The paper introduces an **innovative DPS framework** that integrates tree search and consistency-based sample selection for cost-effective EMS design." [Reviewers uLSg, hUay, 5Egz, 2pz8]
    * "The proposed method outperforms several baseline methods in the experiments of two **representative** structure design **problems**"; "The authors test their method on two **real-world engineering tasks**: Dual-layer Frequency Selective Surface (DualFSS) and High-gain Antenna (HGA); "**Comprehensive experiments** and a detailed ablation study substantiate the effectiveness of each component in improving performance"; "Sufficient ablation studies"; "Two real-world problems are considered." [Reviewers uLSg, 5Egz, 2pz8, hUay]
    * "The methodology is **clearly explained**, with well-structured presentations and **precise formulations** enhancing reader comprehension"; "The paper provides **detailed descriptions** of the Tree-Search-based Design Space Control (TSS) and Consistency-based Sample Selection (CSS) strategies, indicating a deep level of consideration in the algorithm design by the authors"; "The **motivation is clear**." [Reviewers 2pz8, 5Egz, hUay]
    * "demonstrating the **practicality and broad applicability** of their approach"; "The approach **addresses the critical need** for budget-efficient optimization in EMS design and offers applicability to other resource-constrained fields." [Reviewers 2pz8, 5Egz]


2. **We summarize the main modifications in our revised paper (highlighted in blue).**
    * We refined the method description and clarified the problem definition for better precision and understanding.
    * We added more detailed descriptions of the baseline experiment setup in Appendix B.
    * We updated Figure 4 to make the performance comparison more intuitive and visually clear.
    * We added explanations for some variables to improve readability and revised certain variable names to avoid confusion.


Best regards,

The Authors

---

### Meta-Review · Area_Chair_u4u9 · 2024-12-14

**Metareview:**

(a) Summary:
This paper studies the  electromagnetic structure (EMS) design problems by introducing a novel Deep Progressive Search (DPS) method. The authors formulate EMS design as an expensive combinatorial optimization problem. The proposed DPS incorporates a Tree-Search-based Design Space Control (TSS) strategy and a Consistency-based Sample Selection (CSS) mechanism. The approach is evaluated on real-world EMS design tasks, and demonstrates its potential effectiveness.

(b) Strengths:
The use of a Quadtree representation for EMS designs seems novel and promising.
The study considers real-world EMS design problems, which enhances the practical relevance of the proposed approach.

(c) Weaknesses:
The paper lacks comprehensive statistical evaluations, which weakens the reliability of the experimental findings.
Several baselines are outdated, which reduces the credibility of performance comparisons.
Certain aspects of the methodological description are unclear.

(d) Conclusion:
While the paper presents a novel approach with certain practical relevance, the experimental results are not fully convincing due to the absence of robust statistical evaluations and the use of outdated baselines.

**Additional Comments On Reviewer Discussion:**

During the rebuttal, the authors addressed some concerns by making revisions, such as providing more detailed descriptions of the baseline experimental setup and adding explanations for certain variables to improve readability. However, the most critical concerns regarding robust evaluation and the use of stronger baselines remain unresolved.

In the post-rebuttal discussion, three of the four reviewers participated. Two reviewers maintained their recommendation for rejection due to the aforementioned unresolved issues. The third reviewer, while acknowledging these concerns, opted to retain a score of 6, given the novelty of the proposed method. The fourth reviewer, who did not participate in the discussion, kept the initial score of 5.

I concur with the reviewers regarding the unresolved issues, which render this submission below the standard expected for ICLR. Therefore, I recommend rejection.

---

### Decision · Program_Chairs · 2025-01-22

Reject